# Dissecting the heterogeneity of DENV vaccine-elicited cellular immunity using single-cell RNA sequencing and metabolic profiling

Adam T. Waickman[1], Kaitlin Victor[1], Tao Li[1], Kristin Hatch[1], Wiriya Rutvisuttinunt[1], Carey Medin[2], Benjamin Gabriel [2], Richard G. Jarman[1], Heather Friberg [1] & Jeffrey R. Currier[1]

Generating effective and durable T cell immunity is a critical prerequisite for vaccination against dengue virus (DENV) and other viral diseases. However, understanding the molecular mechanisms of vaccine-elicited T cell immunity remains a critical knowledge gap in vaccinology. In this study, we utilize single-cell RNA sequencing (scRNAseq) and longitudinal TCR clonotype analysis to identify a unique transcriptional signature present in acutely activated and clonally-expanded T cells that become committed to the memory repertoire. This effector/memory-associated transcriptional signature is dominated by a robust metabolic transcriptional program. Based on this transcriptional signature, we are able to define a set of markers that identify the most durable vaccine-reactive memory-precursor CD8[+] T cells. This study illustrates the power of scRNAseq as an analytical tool to assess the molecular mechanisms of host control and vaccine modality in determining the magnitude, diversity and persistence of vaccine-elicited cell-mediated immunity.

---

[1] Viral Diseases Branch, Walter Reed Army Institute of Research, Silver Spring, MD, USA. [2] Department of Cell and Molecular Biology, Institute for Immunology and Informatics, University of Rhode Island, Providence, RI, USA. Correspondence and requests for materials should be addressed to A.T.W. (email: adam.t.waickman.ctr@mail.mil)

A fundamental goal of dengue virus (DENV) vaccine design is to generate an immune response that encompasses both humoral and cellular immunity. Consisting of four immunologically and genetically distinct serotypes, DENV-1 to DENV-4, DENV infects up to 280–500 million individuals yearly worldwide[1–3]. While the majority of DENV-infected individuals recover quickly, nearly 500,000 individuals a year develop severe dengue disease, classified as either Dengue Hemorrhagic Fever (DHF) or Dengue Shock Syndrome (DSS)[1–3]. Characterized by increased vascular permeability, hypovolemia, and dysregulated blood clotting, DHF/DSS has a 2.5% mortality rate[1–3]. While the environmental and genetic factors responsible for the development of DHF/DSS are complex and incompletely understood[2–7], prior infection with one serotype of DENV has been shown to significantly increase the likelihood of developing DHF/DSS upon heterotypic re-infection[4,8]. This phenomenon is thought to be facilitated at least in part by poorly-neutralizing, serotype cross-reactive antibodies which enable the opsonization of viral particles but not functional neutralization[2,4,6,9]. Hence, the development of an efficacious DENV vaccine has been significantly complicated by the necessity of generating a protective immune response against four distinct serotypes of DENV, while simultaneously avoiding the development of DHF/DSS potentiated via immune-mediated enhancement of infection. A durable and effective cytotoxic T lymphocyte (CTL) response is considered an important component of DENV immunity that may counteract any deleterious effects of cross-reactive antibodies[10–12].

Generating an effective and durable CTL response by vaccination in humans has been largely attributable to live-attenuated vaccines (e.g., vaccinia, yellow fever) and non-replicating viral vectors (rAD/MVA)[13–16]. Despite the wealth of immune monitoring data that has been generated in recent years, a knowledge gap still exists in our understanding of what determines the magnitude, diversity, and persistence of vaccine-elicited CTL responses in humans. Early studies using systems biology approaches have focused upon innate immune signatures that correlate with adaptive CD8+ T cell responses in following vaccination[17–20]. While these studies have revealed distinct early transcriptional signatures that correlate with the presence of long-term T cell responses, each has relied upon studying sorted T cells at the population level[17–20]. An unknown factor is whether antigen-specific T cells that expand rapidly in response to vaccination within an individual have differential transcriptional profiles that can predict the fate of an individual cell.

TAK-003 is a recombinant, tetravalent DENV vaccine platform derived from the PDK-53 DENV-2 virus strain[21–26]. Recombinant viruses were created using the PDK-53 genetic backbone and the precursor membrane (prM) and envelop (E) genes from DENV-1, −3, and −4, resulting in a vaccine product capable of generating an immune response against all four DENV serotypes[27]. This tetravalent formulation has also been shown to be safe, immunogenic, and capable of providing protective immunity against DENV challenge in both small animal models and non-human primates[21,27,28].

In this study, we demonstrate that TAK-003 elicits a potent cellular immune response which persists for at least 120 days post-vaccination in human subjects. The antigen specificity of the cellular immune response generated by TAK-003 spans the DENV proteome and demonstrates significant cross-reactivity against all four DENV serotypes. Single-cell RNA sequencing analysis of CD8+ T cells activated in response to TAK-003 exposure revealed a highly polyclonal CD8+ T cell repertoire, which had significant clonal overlap between DENV-2 non-structural protein (NS)1- and NS3-reactive CD8+ T cells identified and isolated 120 days post-vaccination. Transcriptional analysis of CD8+ T cells acutely activated in response to

TAK-003 exposure also revealed a highly diverse transcriptional profile, with NS1- and NS3- reactive memory-precursor CD8+ T cells at day 14 post-immunization displaying a distinct transcriptional signature dominated by metabolic pathways. Based on these observations, we identified a panel of metabolic markers, which could be used to faithfully identify CD8+ T cells activated in vitro in response to antigenic stimulation, or activated in vivo in response to TAK-003. In particular, expression of the transferrin receptor (TfR1/CD71)—critical for efficient iron uptake—exclusively marks CD8+ T cells with high proliferative and effector/memory potential. Therefore, analysis of the metabolic profile in vaccine-responsive CD8+ T cells can aid in the identification and characterization of the most effective and durable vaccine-elicited clonotypes.

## Results

**TAK-003 generates potent and durable cellular immunity**. T cell activation in response to TAK-003 administration was assessed by flow cytometry in 55 individuals on days 0, 14, 28, and 120 post immunization. Consistent with other live-attenuated vaccine platforms, TAK-003 administration resulted in significant CD8+ T cell activation on days 14 and 28 post-vaccination (Fig. 1a, b, Supplementary Fig. 1A). CD8+ T cell activation, as assessed by CD38/HLA-DR upregulation, peaked on day 28 post-vaccination and returned to baseline levels by day 120. Moderate CD4+ T cell activation was also observed in response to TAK-003 administration (Fig. 1c, d, Supplementary Fig. 1), with the peak of activation observed on day 14 post-vaccination. However, despite significant activation at this time point, CD8+ T cells do not appear to be functionally licensed to produce IFN-γ in response to DENV-peptide stimulation until later post-vaccination. (Fig. 1e, f, Supplementary Fig. 1B).

To determine if the extensive T cell activation observed in response to TAK-003 administration translated to durable cellular immunity, we stimulated PBMCs isolated from study participants on days 0 and 120 of the study with peptide pools corresponding to the NS1, NS3, NS5, and CprM/E proteins of DENV-1 to -4 and quantified the number of vaccine-reactive, IFN-γ producing cells by ELISPOT assay. All subjects receiving TAK-003 displayed a significant increase in the number of circulating peptide-pool-reactive T cells on day 120 relative to baseline (Supplementary Fig. 1C), with the specificity of this reaction spread across the DENV proteome (Supplementary Fig. 1D). Tetravalent T cell reactivity patterns observed in TAK-003 recipients were detected in both structural (CprM/E) and non-structural (NS1, NS3, NS5) regions of the proteome. Structural region responses could have been generated by any, or all, of the four components of the vaccine, however, non-structural responses can be interpreted as truly cross-reactive since DENV-2 is the common non-structural element of the four vaccine components.

**Identification of antigen-specific CD8+ memory T cells**. To further assess the diversity and persistence of TAK-003-elicited CD8+ T cell immunity, we utilized single-cell RNA sequencing to track the clonal expansion/contraction of TAK-003-reactive CD8+ T cells from acute activation time points and memory time points from the same individual. For this analysis, we utilized samples from a subject who demonstrated strong NS1 and NS3 biased memory T cell responses following TAK-003 administration, as quantified by IFN-γ ELISPOT analysis of day 120 post vaccination PBMC samples (Fig. 2a). Matrixed-based ELISPOT analysis utilizing overlapping peptide pools spanning the entirety of DENV2 NS1 and NS3 revealed that the observed NS1 and NS3

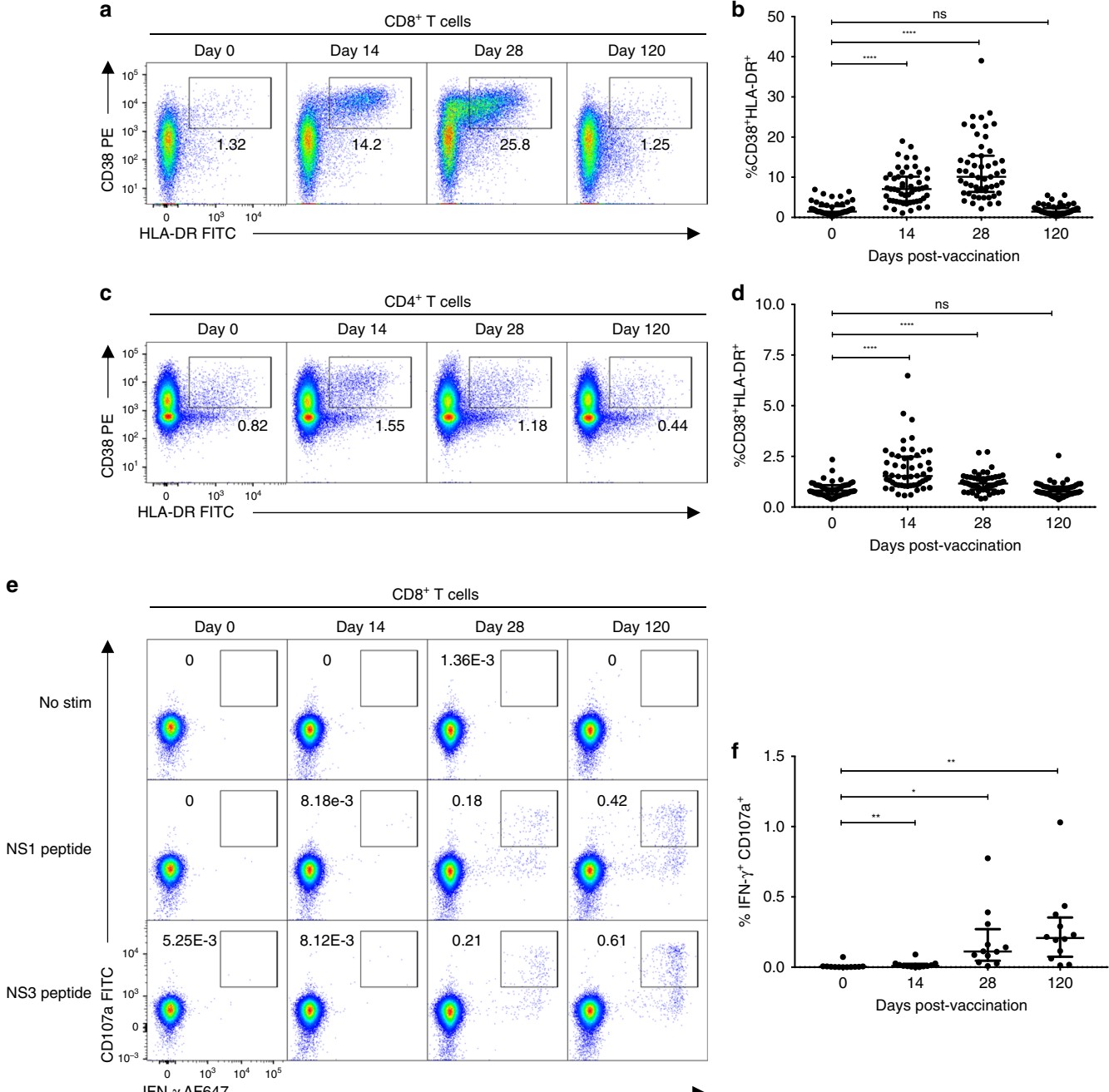

**Fig. 1** TAK-003 elicits a potent and durable DENV-specific cellular immune response following vaccination. PBMCs from individuals immunized with TAK-003 were assessed for markers of T cell activation by flow cytometry on days 0, 14, 28, and 120 post-vaccination. **a** Representative plots demonstrating CD8+ T cell activation as assessed by CD38 and HLA-DR upregulation on days 0, 14, 28, and 120 post-vaccination. **b** Aggregate analysis from 55 TAK-003 recipients, demonstrating maximal CD8+ T cell activation on day 28 post-vaccination, returning to baseline by day 120. **c** Representative plots demonstrating CD4+ T cell activation as assessed by CD38 and HLA-DR upregulation on days 0, 14, 28, and 120 post-vaccination. **d** Aggregate analysis from 55 TAK-003 recipients, demonstrating maximal CD4+ T cell activation on day 14 post-vaccination, returning to baseline by day 120. **e** Representative plots demonstrating DENV-specific IFN-γ production and CD8 T cell degranulation (as assessed by surface CD107a expression) days 0, 14, 28 and 120 post-vaccination. **f** Aggregate analysis from 12 TAK-003 recipients. Cytokine production was assessed following stimulation with peptide pools demonstrated by ELISPOT analysis as being immunogenic at day 120 post vaccination in each individual. *$P < 0.05$, **$P < 0.01$, ****$P < 0.0001$ (Paired two-tailed t-test); ns, not significant. Source data are provided as a Source Data file

reactivity was directed against two distinct epitopes in each peptide pool (Supplementary Table 1).

Following overnight stimulation with either the complete DENV2 NS1 or DENV2 NS3 peptide pool, DENV-reactive memory CD8+ T cells (day 120) were identified by their upregulation of the early activation markers CD25 and CD69.

CD8+CD25+CD69+ T cells were isolated by flow cytometric cell sorting (Fig. 2b, Supplementary Fig. 2) and subjected to single-cell RNA sequencing analysis. In addition, vaccine-reactive CD8+ T cells from an early post-vaccination time point (day 14) were identified by expression of the activation markers CD38 and HLA-DR and isolated by flow sorting (Fig. 2c). A similar number

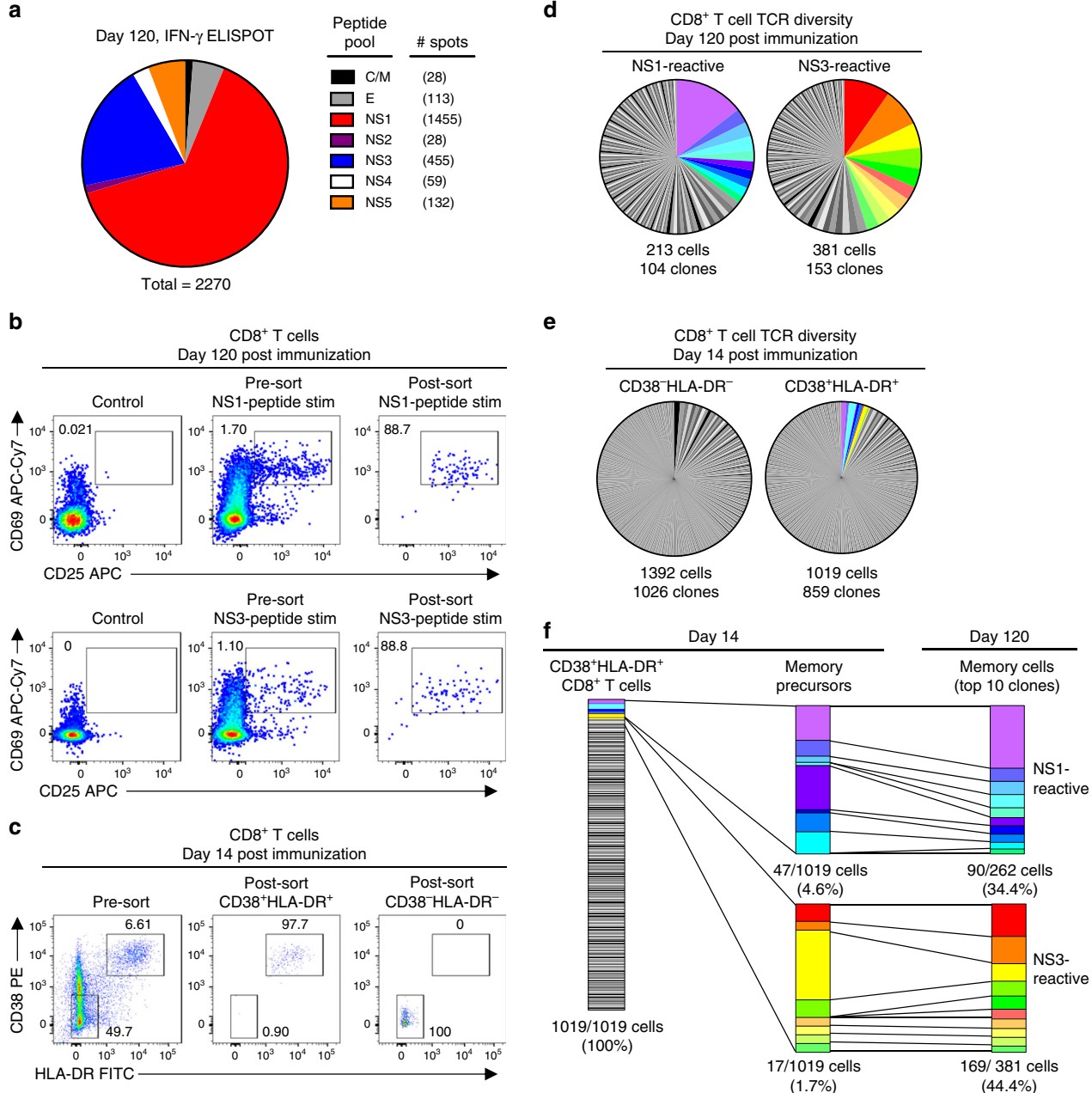

**Fig. 2** TAK-003-elicited CD38+ HLA-DR+ CD8+ T cells are highly polyclonal and persist as long-lived memory cells. TCR clonotype diversity was assessed within acutely-activated and DENV-reactive memory CD8+ T cells using single-cell RNA sequencing. **a** The antigenic specificity of DENV-reactive memory CD8+ T cells from a TAK-003 recipient was assessed by IFN-γ ELISPOT 120 days post immunization. The number of spots is presented relative to 1 million PBMCs. **b** NS1-reactive and NS3-reactive memory CD8+ T cells from 120 post-vaccination were isolated by flow cytometry based on upregulation of CD25 and CD69 expression following in vitro stimulation for 18 h with 1 µg ml⁻¹ of the indicated peptide pools. **c** Activated (CD38+ HLA-DR+) CD8+ T cells and non-activated (CD38−HLA-DR−) CD8+ T cells were additionally isolated by flow cytometry from the same TAK-003 recipient 14 day post-vaccination. **d** TCR clonotype diversity from sorted NS1-reactive and NS3-reactive CD8 + T cells isolated 120 days post-vaccination. The top 10 most abundant NS1- and NS3-reactive clones are demarcated in color. **e** TCR clonotype diversity in sorted non-activated (CD38−HLA-DR−) and activated (CD38+ HLA-DR+) CD8+ T cells using scRNAseq from 14 days post-TAK-003 administration. TCR clones that overlap with the dominant NS1- or NS3-reactive memory CD8+ T cell clones observed at day 120 are indicated with the appropriate color designation. **f** Assessment of the relative contribution and stability of the dominant NS1- and NS3-reactive memory precursors to the overall CD38+HLA-DR+ CD8+ T cell pool at day 14 post-TAK-003 immunization

of CD8+ CD38−HLA-DR− cells (resting) from the same sample were isolated for use as a negative control (Fig. 2c).

A total of 213 NS1-reactive and 381 NS3-reactive CD8+ T cells were identified and isolated from day 120 post-vaccination following in vitro stimulation, yielding 104 and 153 unique TCR clonotypes, respectively from this subject (Fig. 2d, Table 1, Supplementary Fig. 3).

Supplementary Fig. 3). In addition, a total of 1392 CD38−HLA-DR− (control) CD8+ T cells and 1019 CD38+HLA-DR+ (vaccine-reactive) CD8+ T cells were captured in our single-cell RNA sequencing analysis from day 14 post-vaccination, containing 1026 and 859 unique TCR clonotypes, respectively (Fig. 2e, Table 1, Supplementary Fig. 3). The NS1- and NS3- reactive

**Table 1 Dominant TCR clonotypes from acutely activated and memory CD8+T cells**

| Population | Day | Frequency | TCRα CDR3aa | TRAV | TRAJ | TCRβ CDR3aa | TRBV | TRBD | TRB J |
|---|---|---|---|---|---|---|---|---|---|
| CD8+CD38−HLA-DR− (resting) | 14 | 0.011847 | CAVMDSNYQLIW | TRAV1-2 | TRAJ33 | CASSEGAKNIQYF | TRBV6-4 | − | TRBJ2-4 |
| | | 0.009756 | CASMDSNYQLIW | TRAV1-2 | TRAJ33 | CATSDLVQGDTGELFF | TRBV24-1 | TRBD1 | TRBJ2-2 |
| | | 0.008362 | CAMNTDAGGTSYGKLTF | TRAV12-3 | TRAJ52 | CAIRQGNTEAFF | TRBV10-3 | TRBD1 | TRBJ1-1 |
| | | 0.006969 | CAVMDSNYQLIW | TRAV1-2 | TRAJ33 | CSARQGEYEQYF | TRBV20-1 | TRBD1 | TRBJ2-7 |
| | | 0.006272 | CGRRGPPTDKLIF | TRAV13-2 | TRAJ34 | CASSQGPTVGQPQHF | TRBV4-1 | TRBD1 | TRBJ1-5 |
| | | 0.005575 | CAVRDDRGEGTYKYIF | TRAV3 | TRAJ40 | CASSLSQGYQPQHF | TRBV7-6 | TRBD1 | TRBJ1-5 |
| | | 0.004878 | CAVMDSNYQLIW | TRAV1-2 | TRAJ33 | CSARGGLEVDTQYF | TRBV20-1 | TRBD1 | TRBJ2-3 |
| | | 0.004878 | CAVMDSNYQLIW | TRAV1-2 | TRAJ33 | CASSDSGATGELFF | TRBV6-4 | TRBD1 | TRBJ2-2 |
| | | 0.004878 | CAVKDSNYQLIW | TRAV1-2 | TRAJ33 | CSARTGTSVGSFSYEQYF | TRBV20-1 | TRBD2 | TRBJ2-7 |
| | | 0.004181 | CAVLDSNYQLIW | TRAV1-2 | TRAJ33 | CSARDLDRDNSPLHF | TRBV20-1 | TRBD1 | TRBJ1-6 |
| CD8+CD38+HLA-DR+ (activated) | 14 | 0.01271571 | CAVRGRGDYKLSF | TRAV1-2 | TRAJ20 | CASSSAGTLNTGELFF | TRBV7-9 | TRBD1 | TRBJ2-2 |
| | | 0.00999092 | CAGRGAGSYQLTF | TRAV21 | TRAJ28 | CASSLLSYEQYF | TRBV7-9 | − | TRBJ2-7 |
| | | 0.00726612 | CAVQAGGYSTLTF | TRAV20 | TRAJ11 | CASAEADNEQFF | TRBV13 | TRBD2 | TRBJ2-1 |
| | | 0.00726612 | CAVRFPRDYKLSF | TRAV21 | TRAJ20 | CASSPTGTGYYEQYF | TRBV7-9 | TRBD1 | TRBJ2-7 |
| | | 0.00635786 | CAEMEGFKTIF | TRAV5 | TRAJ9 | CASSFLVGPGGNTIYF | TRBV7-9 | − | TRBJ1-3 |
| | | 0.00544959 | CAVRDSRAAGNKLTF | TRAV1-2 | TRAJ17 | CASSKGFTSTDTQYF | TRBV6-4 | − | TRBJ2-3 |
| | | 0.00544959 | CAVRRGGSYIPTF | TRAV21 | TRAJ6 | CAGSGIQGELFF | TRBV19 | TRBD1 | TRBJ2-2 |
| | | 0.00454133 | CALSGINTDKLIF | TRAV16 | TRAJ34 | CASSPKWDGQETQYF | TRBV3-1 | TRBD2 | TRBJ2-5 |
| | | 0.00454133 | CAPLGGAGSYQLTF | TRAV21 | TRAJ28 | CASSPRQGNTGELFF | TRBV7-9 | TRBD1 | TRBJ2-2 |
| | | 0.00454133 | CIVRSLINYGQNFVF | TRAV26-1 | TRAJ26 | CASSSVSYEQYF | TRBV7-9 | − | TRBJ2-7 |
| NS1 reactive memory CD8+ | 120 | 0.173516 | CAGRGAGSYQLTF | TRAV21 | TRAJ28 | CASSLLSYEQYF | TRBV7-9 | − | TRBJ2-7 |
| | | 0.036530 | CAPLGGAGSYQLTF | TRAV21 | TRAJ28 | CASSPRQGNTGELFF | TRBV7-9 | TRBD1 | TRBJ2-2 |
| | | 0.036530 | CAGVTGGSYIPTF | TRAV27 | TRAJ6 | CASSGRAHYGYTF | TRBV7-9 | TRBD1 | TRBJ1-2 |
| | | 0.036530 | CAVGGYNKLIF | TRAV3 | TRAJ4 | CSARASTVPLYEQYF | TRBV20-1 | TRBD1 | TRBJ2-7 |
| | | 0.027397 | CAMREGNTGGFKTIF | TRAV14DV4 | TRAJ9 | CAISESQDNQPQHF | TRBV10-3 | TRBD1 | TRBJ1-5 |
| | | 0.022831 | CAVRGRGDYKLSF | TRAV1-2 | TRAJ20 | CASSSAGTLNTGELFF | TRBV7-9 | TRBD1 | TRBJ2-2 |
| | | 0.022831 | CARSYNTDKLIF | TRAV19 | TRAJ34 | CASSLEIEAFF | TRBV7-9 | − | TRBJ1-1 |
| | | 0.022831 | CAVRRGGSYIPTF | TRAV21 | TRAJ6 | CAGSGIQGELFF | TRBV19 | TRBD1 | TRBJ2-2 |
| | | 0.018265 | CAEMEGFKTIF | TRAV5 | TRAJ9 | CASSFLVGPGGNTIYF | TRBV7-9 | − | TRBJ1-3 |
| | | 0.013699 | CAMIGNTPLVF | TRAV14DV4 | TRAJ29 | CASSGLEAEQFF | TRBV7-9 | − | TRBJ2-1 |
| CD8+ NS3 reactive memory CD8+ | 120 | 0.09438776 | CAVQAQGYSTLTF | TRAV20 | TRAJ11 | CASSELDNEQFF | TRBV2 | − | TRBJ2-1 |
| | | 0.07908163 | CAVGSPLYSGGGADGLTF | TRAV8-3 | TRAJ45 | CASSLAGGYEQYF | TRBV5-6 | TRBD2 | TRBJ2-7 |
| | | 0.05102041 | CAVQAGGYSTLTF | TRAV20 | TRAJ11 | CASAEADNEQFF | TRBV13 | TRBD2 | TRBJ2-1 |
| | | 0.04336735 | CAVNGGVGNQFYF | TRAV12-2 | TRAJ49 | CSTGTPGQPLSYEQYF | TRBV20-1 | TRBD1 | TRBJ2-7 |
| | | 0.03826531 | CAASWYSGGGADGLTF | TRAV29DV5 | TRAJ45 | CASSQSDRSTYNEQFF | TRBV4-3 | − | TRBJ2-1 |
| | | 0.02806122 | CAGTGGADGLTF | TRAV12-2 | TRAJ45 | CASNPGSPEAFF | TRBV2 | TRBD2 | TRBJ1-1 |
| | | 0.02806122 | CAADGGSQGNLIF | TRAV21 | TRAJ42 | CASSEWIGTEAFF | TRBV6-1 | − | TRBJ1-1 |
| | | 0.0255102 | CAVLNSGYSTLTF | TRAV21 | TRAJ11 | CATSGVGDRGFDNEQFF | TRBV24-1 | TRBD1 | TRBJ2-1 |
| | | 0.0255102 | CALDIMDSSYKLIF | TRAV6 | TRAJ12 | CATSDLGLPEETQYF | TRBV24-1 | TRBD2 | TRBJ2-5 |
| | | 0.01785714 | CAVDNNNARLMF | TRAV39 | TRAJ31 | CASSPRFGDTYEQYF | TRBV11-3 | TRBD1 | TRBJ2-7 |

memory CD8+ T cells isolated on day 120 post-vaccination exhibited a significant degree of clonal diversity, with an average (mean) clonal abundance of 1.71 cells/clone for NS1-reactive cells, and 2.49 cells/clone for NS3-reactive cells. However, these statistics are significantly skewed by a large number of clones with a single representative in the dataset. In contrast, the top 10 most abundant clones found in both the NS1- and NS3-reactive memory T cell pool account for 34.4 and 44.4% of recovered cells in the dataset, respectively (Fig. 2d, f). Furthermore, 80% of these dominant NS1- and NS3-reactive memory CD8+ T cell clones identified on day 120 post-vaccination can also be observed in the CD38+HLA-DR+ CD8+ T cell pool isolated on day 14 post-vaccination (Fig. 2f). The dominant NS1- and NS3-reactive memory precursors account for only 6.3% of all CD38+ HLA-DR+ CD8+ T cells isolated on day 14 post-vaccination (NS1-reactive precursors = 4.6% of CD38+HLA-DR+ CD8+ T cells; NS3-reactive precursors = 1.7% of all CD38+HLA-DR+ CD8+ T cells). Minimal overlap was observed between the TCR repertoires of sorted activated CD38+HLA-DR+ CD8+ T cells and resting CD38−HLA-DR− CD8+ T cells on day 14 post vaccination (16 total overlapping clones). Six of these overlapping clonotypes corresponded to the semi-invariant TRAV1-2/TRAJ33 arrangement associated with Mucosal-Associated Invariant T cells (MAITs), which appears to be a substantial population in this subject.

**Characterization of TAK-003-stimulated CD8+ T cells.** In light of the significant clonal diversity observed in the activated CD38+ HLA-DR+CD8+ T cell compartment 14 days after TAK-003

administration, we decided to determine if the heterogeneity of this population extended to the functional transcriptional profile of these cells. To this end, we assessed the functional gene expression profile of the sorted CD8+ CD38+HLA-DR+ T cells isolated 14 days post-TAK-003 inoculation using single-cell RNAseq. Of the 1019 cells contained within the TCR analysis dataset previously described, we recovered a full gene transcriptional profile from 1003 cells (98.4%) that met the quality control threshold of our analysis pipeline.

Unsupervised tSNE clustering of sorted CD38+HLA-DR+ CD8+ T cells based on differential gene expression profiles revealed four statistically-distinct populations (Fig. 3a, b, Table 2). Expression of genes associated with an effector CD8+ T cell program such as GZMA, GZMB, and PRF1 were significantly enriched in cluster 1 (Fig. 3b, c, Table 2). Transcripts associated with a more resting/naïve T cell phenotype such as CCR7, MAL, and TCF7 were significantly enriched in clusters 2 and 3, and showed minimal overlap with cells expressing effector gene products (Fig. 3b, c, Table 2). In addition, gene expression associated with cellular metabolism, proliferation and cell-cycle progression, such as TYMS, IDH2, and GAPDH were significantly enriched in cluster 1 relative to all other groups.

To compare how the transcriptional profile of phenotypically activated CD38+HLA-DR+ CD8+ T cells compared to that of phenotypically non-activated CD38−HLA-DR− CD8+ T cells, we performed additional single cell gene expression analysis on the sorted CD38−HLA-DR− CD8+ T cells shown Fig. 2c. The analysis resulted in the identification of 1391 cells with a complete gene expression profile along with a full-length TCR. This

scRNAseq gene expression data was merged with the activated CD38$^+$HLA-DR$^+$ CD8$^+$ T cell gene expression data, and differential gene expression analysis performed (Supplementary Fig. 4A, Supplementary Table 2). Notably, there was minimal transcriptional overlap between the sorted CD38$^+$HLA-DR$^+$ and CD38$^-$HLA-DR$^-$ CD8$^+$ T cells, with each statistically defined cluster in the merged dataset exhibiting significant bias towards one-or-the-other parental population (Supplementary Fig. 4A, Supplementary Fig. 4B). Canonical T cell activation markers (HLA-DRA, CD27, GZMH) and genes associated with cellular proliferation/migration (STMN1, HMGB2, TUBB) are preferentially expressed in phenotypically activated CD8$^+$ T cells (Supplementary Fig. 4A, C, Supplementary Table 2). Interestingly for this subject, a statistically unique transcriptional cluster can be defined within the merged dataset (cluster 1) that is significantly enriched in CD38$^-$HLA-DR$^-$ CD8$^+$ T cells and characterized by expression of the canonical MAIT-associated TRAV1-2 TCR receptor gene segment[29,30], along with the key linage-defining genes KLRB1, KLRG1 and IL-7R (Supplementary Fig. 4C, Supplementary Table 2)[31,32]. The prevalence of CD8$^+$ MAIT cells within the sorted CD38$^-$HLA-DR$^-$ CD8$^+$ T cell population is further validated by the corresponding TCR clonotype information derived from these cells (Table 1), which shows a significant enrichment in cells expressing the canonical TRAV1-2 TRAJ33 semi-invariant TCR alpha chain[29,30]. These data demonstrate that although there is significant transcriptional heterogeneity within the sorted CD38$^+$HLA-DR$^+$ CD8$^+$ T cells circulating after TAK-003 administration, this population is transcriptionally distinct from phenotypically non-activated CD38$^-$HLA-DR$^-$CD8$^+$ T cells.

Having demonstrated that CD38$^+$HLA-DR$^+$ CD8$^+$ T cells are a distinct yet transcriptionally heterogeneous population 14 days post TAK-003 administration, we asked if DENV-reactive CD8$^+$ T cells that are destined to form long-lived memory T cells can be identified within the population, and if they exhibit a unique transcriptional profile relative to all other activated CD8$^+$ T cells. To this end, we asked whether any of the TCR clonotypes defined in the NS1$^-$ or NS3- reactive memory CD8$^+$ T cell population 120 days post TAK-003 administration could be found amongst CD38$^+$HLA-DR$^+$CD8$^+$T cells on day 14 post vaccination.

Of the 1003 CD38$^+$HLA-DR$^+$ CD8$^+$ T cells recovered on day 14 post-vaccination, 145 cells (14.5%) expressed TCRs found in NS1- and NS3-reactive memory cells on day 120 post-vaccination (Fig. 3d). The transcriptional profile of these memory precursors positioned them predominantly within the previously defined phenotypic cluster 1 (Supplementary Fig. 5), suggesting that this distinct subset of CD38$^+$HLA-DR$^+$ CD8$^+$ T cells found 14 days post-vaccination is uniquely primed to develop into long-lived memory cells. In addition, those cells expressing a clonally expanded TCR (defined as a TCR expressed in >2 cells) were preferentially enriched in cluster 1 (Fig. 3e).

To further define the transcriptional and phenotypic signatures of vaccine-reactive CD8$^+$ T cells within our dataset, we utilized the Ingenuity Pathway Analysis (IPA) software package[33] to identify gene pathways selectively expressed in putative DENV-reactive effector/memory-precursor CD8$^+$ T cells (Table 3). We assessed the gene pathways preferentially expressed within cluster 1, which contained the majority of identified memory-precursor cells, relative to all other cells in the dataset. Interestingly, there was some preferential expression of gene pathways associated with effector function and cellular migration in cluster 1. However, the dominant cellular transcriptional signatures that distinguished cluster 1 from the rest of the dataset were associated with cellular metabolism and proliferation, such as oxidative phosphorylation, mTOR signaling, and eIF4/p70S6K signaling (Table 3). These data

suggest that the assessment of cellular metabolism pathways may provide a robust and unbiased indication of cellular memory-precursor potential, as well as effector status and antigen reactivity.

**Polyclonal T cell activation modulates cellular metabolism.** To explore the broader implications of the observed relationship between metabolic gene activity and effector/memory potential in vaccine-reactive T cells, we aimed to establish a set of markers to quantify the metabolic potential of human T cells following TCR engagement. As the metabolically-associated gene products identified by scRNAseq analysis and preferentially expressed in memory-precursor CD8$^+$ T cells were intracellular in origin, we selected a panel of cell surface markers and traceable metabolites that we hypothesized would be preferentially expressed in, or taken up by, memory-precursor CD8$^+$ T cells. The criteria used to select these markers were that they (1) were directly regulated by the gene pathways differentially expressed in our populations of interest and/or (2) the associated metabolites were utilized by metabolic pathways differentially expressed in our populations of interest.

To this end, we selected the transferrin receptor complex (TfR1/ CD71) and the Large-neutral Amino Acid Transporter 1 (LAT1/ CD98) for additional analysis. The expression and surface-localization of both of these transporters is regulated by mTOR/ p70S6K signaling (pathways preferentially expressed in our memory-precursor cells)[34–36]. Furthermore, the metabolites they import—iron and amino acids, respectively—are critical co-factors for mitochondrial oxidative phosphorylation or direct catabolic and anabolic substrates required in proliferating T cells, which can both directly impact mTOR/p70S6K signaling[37–39]. In addition, we assessed the ability of T cells to uptake glucose (measured by uptake of 2-NBDG, a fluorescent analog of glucose)[40,41] or fatty acids (assessed by BODIPY FL-C$_{16}$ uptake, a florescent palmitate derivative)[42,43] by flow cytometry.

To assess the suitability of these markers to quantify the metabolic potential of human T cells following TCR engagement, we stimulated PBMCs from normal healthy donors in vitro with αCD3/CD28 to induce polyclonal T cell activation. After in vitro stimulation, CD4$^+$ and CD8$^+$ T cells showed a well-characterized pattern of activation marker upregulation (Fig. 4a, b, Supplementary Fig. 6). Furthermore, we observed a sustained increase in expression of TfR1 and CD98 following TCR stimulation (Fig. 4c, d, Supplementary Fig. 6), as well as 2-NBDG and BODIPY FL-C$_{16}$ uptake (Fig. 4e, f, Supplementary Fig. 6). Uptake of 2-NBDG and BODIPY FL-C$_{16}$ by activated T cells could be reduced upon addition of the corresponding unlabeled metabolites (Supplementary Fig. 7A), although the inhibition of 2-NBDG uptake by unlabeled glucose was modest. Of particular note, the expression of TfR1 and CD98 and uptake of BODIPY FL-C$_{16}$ showed exceptional promise as markers of T cell activation and metabolic potential due to their large dynamic range and persistence relative to other markers of T cell activation. In particular, the expression of TfR1 on in vitro stimulated CD8$^+$ T cells increased ~600 fold 48 h after polyclonal stimulation. The increase in TfR1 expression upon in vitro stimulation corresponded to a significant increase in the ability of T cells to uptake transferrin (Supplementary Fig. 7B).

**Clonal T cell activation modulates cellular metabolism.** In light of the observation that the expression of TfR1 and CD98 and the uptake of BODIPY FL-C$_{16}$ are quantifiable metabolic indicators of polyclonal T cell activation following in vitro activation, we endeavored to determine if these markers could be utilized to identify and characterize T cells activated in an antigen-specific

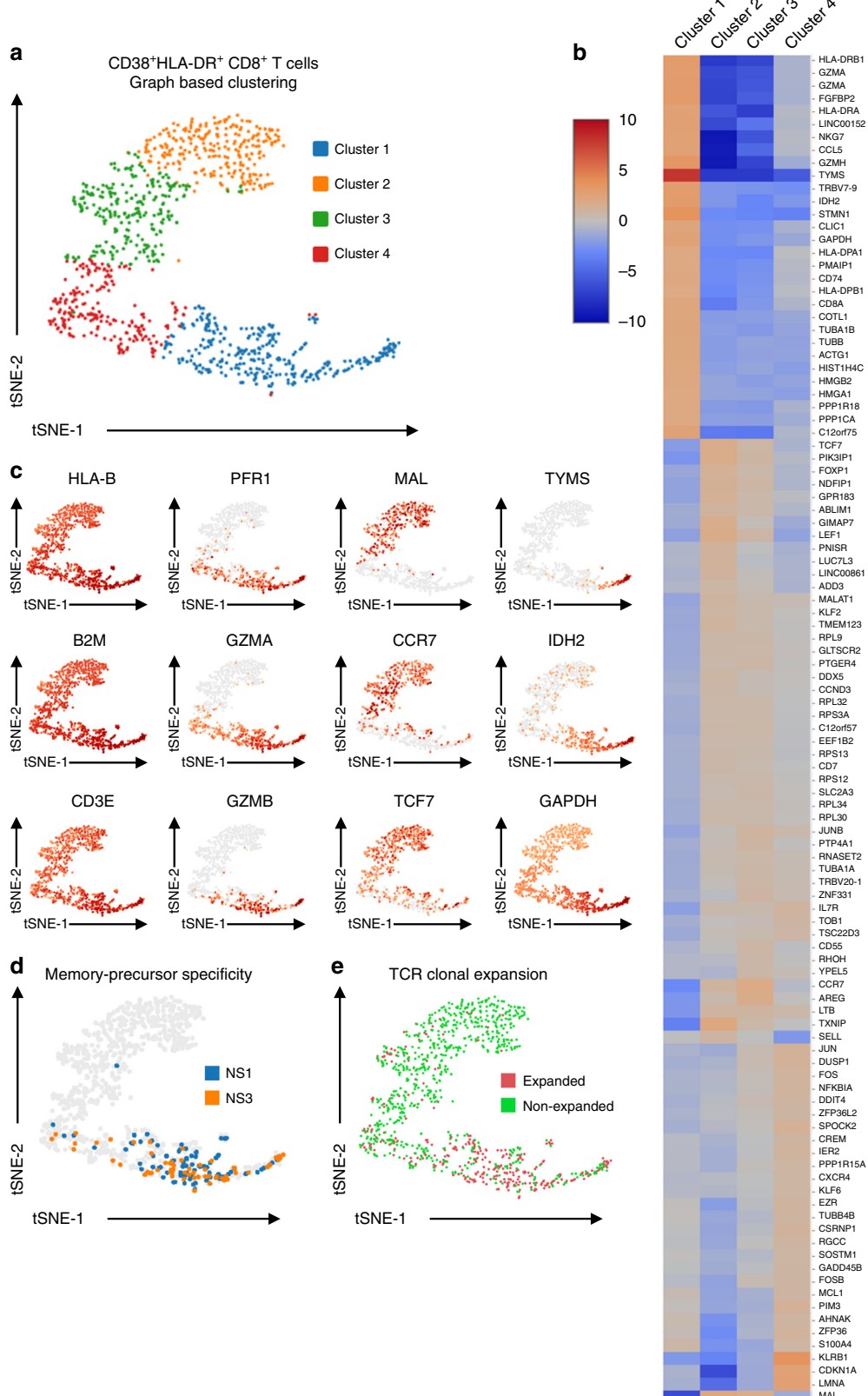

**Fig. 3** TAK-003-elicited CD8+ memory precursors exhibit a unique transcriptional profile. **a** Transcriptional heterogeneity of sorted CD38+HLA-DR+ CD8+ T cells isolated 14 days post-TAK-003 administration as assessed by single-cell RNA sequencing. Unsupervised cell clustering and data visualization were performed using sparse nearest-neighbour graphing, followed by Louvain Modularity Optimization. **b** Hierarchical heat map of differentially expressed genes within the sorted CD38+HLA-DR+ CD8 T cell. **c** Expression of select transcripts within the sorted CD38+HLA-DR+ CD8+ T cells. **d** Distribution of CD38+HLA-DR+ CD8+ T cells at day 14 with TCR clonotypes overlapping with NS1- and NS3-reactive memory CD8+ T cells at day 120 (memory precursors). **e** Distribution of clonally expanded CD38+HLA-DR+ CD8+ T cells at day 14

**Table 2 Cluster defining gene lists from day 14 sorted CD38+HLA-DR+ CD8+ T cell scRNAseq analysis**

| Cluster 1 | | | | Cluster 2 | | | | Cluster 3 | | | | Cluster 4 | | | |
|---|---|---|---|---|---|---|---|---|---|---|---|---|---|---|---|
| Ensembl ID | Gene Name | P-Value | Log2 FC | Ensembl ID | Gene Name | P-Value | Log2 FC | Ensembl ID | Gene Name | p-Value | Log2 FC | Ensembl ID | Gene Name | P-Value | Log2 FC |
| ENSG00000176890 | TYMS | 4.53E-33 | 8.03 | ENSG00000265972 | TXNIP | 4.93E-13 | 2.21 | ENSG00000126353 | CCR7 | 5.01E-08 | 1.94 | ENSG00000111796 | KLRB1 | 1.51E-22 | 3.77 |
| ENSG00000100450 | GZMH | 8.67E-29 | 3.54 | ENSG00000172005 | MAL | 1.05E-10 | 2.25 | ENSG00000109321 | AREG | 2.19E-05 | 1.83 | ENSG00000160789 | LMNA | 1.16E-11 | 2.77 |
| ENSG00000182054 | IDH2 | 1.58E-25 | 3.15 | ENSG00000100100 | PIK3IP1 | 6.62E-08 | 1.79 | ENSG00000172005 | MAL | 2.26E-05 | 1.68 | ENSG00000124762 | CDKN1A | 1.29E-09 | 2.72 |
| ENSG00000145649 | GZMA | 5.86E-25 | 3.10 | ENSG00000138795 | LEF1 | 1.46E-07 | 1.83 | ENSG00000171223 | JUNB | 9.93E-03 | 1.07 | ENSG00000107742 | SPOCK2 | 6.97E-04 | 1.39 |
| ENSG00000117632 | STMN1 | 5.84E-24 | 3.80 | ENSG00000081059 | TCF7 | 2.25E-07 | 1.73 | ENSG00000227507 | LTB | 2.98E-02 | 0.98 | ENSG00000120129 | DUSP1 | 2.11E-03 | 1.30 |
| ENSG00000196126 | HLA-DRB1 | 7.29E-21 | 3.18 | ENSG00000179144 | GIMAP7 | 6.65E-07 | 1.69 | ENSG00000130844 | ZNF331 | 3.44E-02 | 0.98 | ENSG00000177606 | JUN | 3.66E-03 | 1.30 |
| ENSG00000137441 | FGFBP2 | 1.83E-20 | 3.19 | ENSG00000131507 | NDFIP1 | 1.02E-04 | 1.36 | ENSG00000131507 | NDFIP1 | 3.85E-02 | 0.94 | ENSG00000198355 | PIM3 | 4.45E-03 | 1.32 |
| ENSG00000222041 | LINC00152 | 2.24E-19 | 2.78 | ENSG00000114861 | FOXP1 | 2.21E-04 | 1.34 | ENSG00000100100 | PIK3IP1 | 3.90E-02 | 0.96 | ENSG00000100906 | NFKBIA | 1.23E-02 | 1.13 |
| ENSG00000235162 | C12orf75 | 4.16E-19 | 2.64 | ENSG00000169508 | GPR183 | 2.81E-04 | 1.39 | ENSG00000112245 | PTP4A1 | 4.44E-02 | 0.96 | ENSG00000160888 | IER2 | 1.31E-02 | 1.12 |
| ENSG00000213719 | CLIC1 | 8.58E-18 | 2.39 | ENSG00000132424 | PNISR | 1.18E-03 | 1.19 | ENSG00000128016 | CD55 | 7.50E-02* | 0.88 | ENSG00000128016 | ZFP36 | 1.31E-02 | 1.15 |
| ENSG00000100453 | GZMB | 9.00E-18 | 3.16 | ENSG00000100848 | LUC7L3 | 1.74E-03 | 1.21 | ENSG00000168685 | PTGER4 | 7.68E-02* | 0.87 | ENSG00000168685 | IL7R | 1.40E-02 | 1.12 |
| ENSG00000103187 | COTL1 | 1.10E-16 | 2.27 | ENSG00000152558 | TMEM123 | 4.46E-03 | 1.09 | ENSG00000081059 | TCF7 | 7.89E-02* | 0.88 | ENSG00000144655 | CSRNP1 | 1.45E-02 | 1.17 |
| ENSG00000153563 | CD8A | 1.35E-16 | 2.34 | ENSG00000245164 | LINC00861 | 5.02E-03 | 1.15 | ENSG00000119801 | YPEL5 | 9.27E-02* | 0.81 | ENSG00000092820 | EZR | 2.44E-02 | 1.08 |
| ENSG00000204287 | HLA-DRA | 3.1IE-14 | 2.75 | ENSG00000227507 | LTB | 8.35E-03 | 1.04 | ENSG00000163682 | RPL9 | 9.60E-02* | 0.80 | ENSG00000170345 | FOS | 3.53E-02 | 1.06 |
| ENSG00000019582 | CD74 | 4.1IE-14 | 2.13 | ENSG00000251562 | MALAT1 | 8.82E-03 | 0.99 | ENSG00000138795 | LEF1 | 1.13E-01* | 0.86 | ENSG00000152518 | ZFP36L2 | 6.28E-02* | 0.96 |
| ENSG00000111640 | GAPDH | 4.1IE-14 | 2.61 | ENSG00000099204 | ABLIM1 | 9.1IE-03 | 1.05 | ENSG00000167552 | TUBA1A | 1.15E-01* | 0.80 | ENSG00000088229 | TUBB4B | 7.09E-02* | 1.02 |
| ENSG00000271503 | CCL5 | 4.1IE-14 | 2.73 | ENSG00000126353 | CCR7 | 1.27E-02 | 1.06 | ENSG00000168421 | RHOH | 1.29E-01* | 0.79 | ENSG00000027760 | RGCC | 7.12E-02* | 0.99 |
| ENSG00000184009 | ACTG1 | 4.1IE-14 | 2.28 | ENSG00000188404 | SELL | 1.42E-02 | 0.98 | ENSG00000111678 | C12orf57 | 1.42E-01* | 0.75 | ENSG00000143384 | MCL1 | 7.30E-02* | 0.98 |
| ENSG00000105374 | NKG7 | 4.1IE-14 | 2.71 | ENSG00000127528 | KLF2 | 1.88E-02 | 0.94 | ENSG00000059804 | SLC2A3 | 1.48E-01* | 0.75 | ENSG00000168209 | DDIT4 | 7.39E-02* | 0.99 |
| ENSG00000172531 | PPP1CA | 4.96E-14 | 2.05 | ENSG00000108654 | DDX5 | 3.22E-02 | 0.88 | ENSG00000144713 | RPL32 | 1.57E-01* | 0.72 | ENSG00000095794 | CREM | 8.85E-02* | 0.95 |

*not significant

fashion. Therefore, we stimulated PBMCs from healthy donors with peptide pools corresponding to the proteomes of common viral pathogens (hCMV, HBV, adenovirus, and influenza), and assessed the upregulation of conventional activation markers (CD69, CD25), metabolite transporters (TfR1, CD98), as well as the utilization of metabolic substrates (BODIPY FL-C$_{16}$) by flow cytometry.

We observed significant donor-to-donor variability in the fraction of CD8$^+$ T cells responding to virus-antigen stimulation (Fig. 5a, b), with the most abundant CD8$^+$ T cell activation occurring in response to CMV peptide stimulation. Moderate activation was observed in response to adenovirus and influenza, while minimal activation was seen in response to HBV peptide stimulation. In addition to upregulating expression of CD69 and CD25 following in vitro antigenic stimulation, we observed that CD8$^+$ T cells concurrently increased their uptake of BODIPY FL-C$_{16}$ (Fig. 5a, c), expression of CD98 (Fig. 5a, d), as well as expression of TfR1 (Fig. 5a, e). There was a significant degree of correlation between the upregulation of the conventional activation makers CD25/CD69 following in vitro stimulation and the increased expression of TfR1, CD98, or the uptake of BODIPY FL-C$_{16}$ following hCMV (Fig. 5f), adenovirus (Fig. 5g), or influenza (Fig. 5h) peptide stimulation in these same cells. These data demonstrate that the antigen-specific activation of clonal CD8$^+$ T cells can be quantified and functionally characterized using a combination of conventional activation markers, metabolic transporters, and fluorescent metabolic substrates.

**In vivo T cell activation modulates cellular metabolism.** Having established the utility of TfR1/CD98 expression, as well as BODIPY FL-C$_{16}$ and 2-NBDG uptake, for the identification and functional characterization of antigen-specific in vitro activated T cells, we aimed to determine whether these same markers can be used to dissect the functional heterogeneity of in vivo activated, vaccine-reactive T cells. To this end, we compared the expression of the conventional activation markers CD38/HLA-DR on T cells isolated from individuals inoculated with TAK-003 on days 0, 14, 28 and 120 post-immunization, to the ability of cells to uptake 2-NBDG and BODIPY FL-C$_{16}$, and the upregulation of the metabolic transporters TfR1 and CD98.

As previously shown, TAK-003 administration elicits a potent CD8$^+$ T cell response as assessed by CD38/HLA-DR expression, with the peak of CD8$^+$ T cell expansion observed on day 28 post-inoculation (Fig. 6a, Supplementary Fig. 8). A similar, albeit reduced, pattern of CD4$^+$ T cell activation can be concurrently observed in vaccinated individuals, with the peak of CD4$^+$ T cell expansion occurring 14 days post-inoculation (Supplementary Fig. 9).

Consistent with the single-cell RNA sequencing phenotype observed in TAK-003-reactive T cells and our in vitro T cell stimulation data, a significant increase in both 2-NBDG and BODIPY FL-C$_{16}$ uptake can be observed in HLA-DR positive CD8$^+$ (Fig. 6b, c) and CD4$^+$ (Supplementary Fig. 9) T cells on days 14 and 28 post-vaccination. Additionally, a subset of HLA-DR positive CD8$^+$ and CD4$^+$ T cells upregulate TfR1 (Fig. 6d, Supplementary Fig. 9) and CD98 (Fig. 6e, Supplementary Fig. 9) on days 14 and 28 post-TAK-003 administration. Notably, while the peak expression of CD38/HLA-DR on CD8$^+$ T cells occurs on day 28 post-vaccination, the expression of both TfR1 and CD98 peaks on day 14, suggesting that these markers may provide an earlier indicator of vaccine T cell immunogenicity than conventional surface markers. The expression of the conventional activation markers CD38 and HLA-DR and the uptake of BODIPY FL-C$_{16}$ and 2-NBDG appear to occur concomitantly and uniformly in in vivo activated T cells whereas

**Table 3 Enriched gene pathways in memory-precursor CD8$^+$ T cells**

| Pathway | P-value | z score | Ratio | Major contributing genes |
|---|---|---|---|---|
| Oxidative phosphorylation | 1.00E-36 | 4.667 | 0.346 | NDUFB9, COX7B, MT-ND5, UQCR10, ATP5PB |
| EIF2 signaling | 1.00E-32 | −4.146 | 0.202 | RPS26,RPL32, RPS27, EIF4A2, RPL35A, RPS5 |
| Mitochondrial dysfunction | 3.16E-31 | – | 0.23 | NDUFB9, COX7B, MT-ND5, UQCR10, ATP5PB |
| mTOR signaling | 2.51E-16 | 0.816 | 0.137 | RPS26, RPS27, FKBP1A, EIF4B, EIF4A2, RPS5 |
| Regulation of eIF4 and p70S6K signaling | 6.31E-15 | – | 0.149 | RPS29, RPS6, RPS26, RPS27, RPS18, PABPC1 |
| Sirtuin signaling pathway | 6.31E-15 | 0.654 | 0.106 | JUN, NDUFB9, MT-ND5, ATP5PB, PGAM1 |
| Regulation of Actin-based motility by Rho | 2.00E-13 | 4 | 0.198 | ARPC3, MYL12B, WIPF1, ARPC5L, RAC2, ACTR3 |
| Signaling by Rho Family GTPases | 5.01E-12 | 2.836 | 0.099 | JUN, WIPF1, ARPC5L, ARPC1B, FOS, ARPC5 |
| RhoGDI signaling | 7.94E-12 | −3.299 | 0.119 | ARPC3, MYL12B, MSN, ARPC5L, ARPC1B |
| Cdc42 signaling | 2.00E-11 | 2.13 | 0.14 | JUN, HLA-DPB1, ARPC3, MYL12B, WIPF1 |

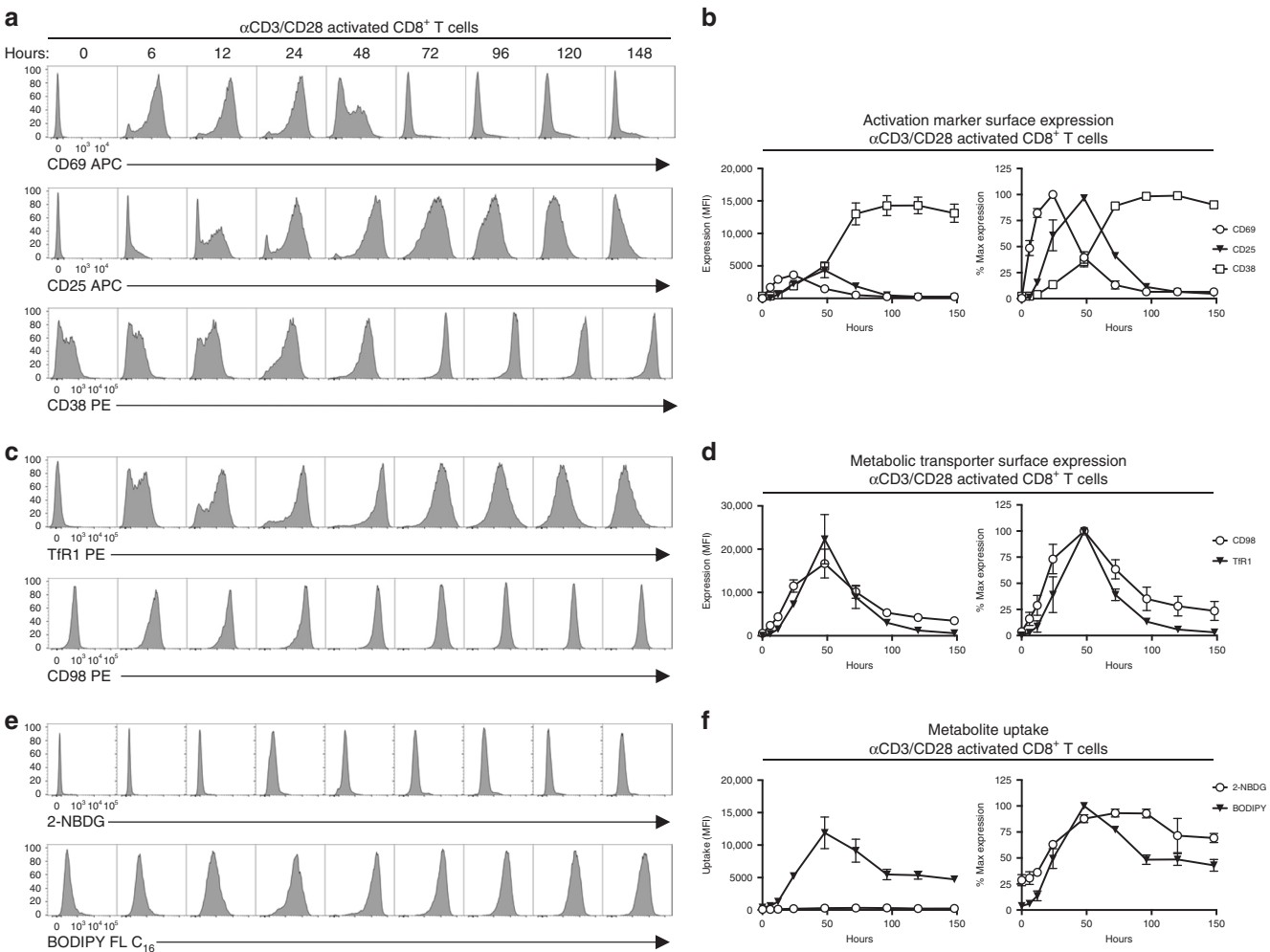

**Fig. 4** In vitro activated T cells can be identified by changes in metabolite transporter expression and metabolite utilization. CD8$^+$ T cells from healthy donors were analyzed by flow cytometry at the indicated time points after in vitro stimulation with 0.1 μg ml$^{-1}$ αCD3 and 1 μg ml$^{-1}$ αCD28. T cell activation was assessed based on expression of **a**, **b** CD69, CD25, **c**, **d** TfR1, CD98, and **e**, **f** 2-NBDG BODIPY FL-C16 uptake. Error bars show mean and SEM. Results are representative of two independent experiments with a total of four individual donors. Source data are provided as a Source Data file

the expression of TfR1 (transferrin receptor) shows a significant amount of variability within the HLA-DR$^+$ T cell compartment (Fig. 6d). As quantification of CD38, TfR1, and CD98 was all performed using antibodies conjugated to the same fluorophore (PE), the observed heterogeneity in TfR1 expression within the HLA-DR$^+$ T cell compartment is unlikely to be attributable to a technical aberration. This suggests the possibility that heterogeneity of TfR1 expression may reflect and capture the functional

diversity previously observed in the scRNAseq-derived transcriptional profiles of vaccine-reactive CD8$^+$ T cells, and can be utilized to more stringently identify effector/memory-precursor CD8$^+$ T cells.

**TfR1 expression correlates with effector/memory potential**. To investigate whether differences in the surface expression of TfR1

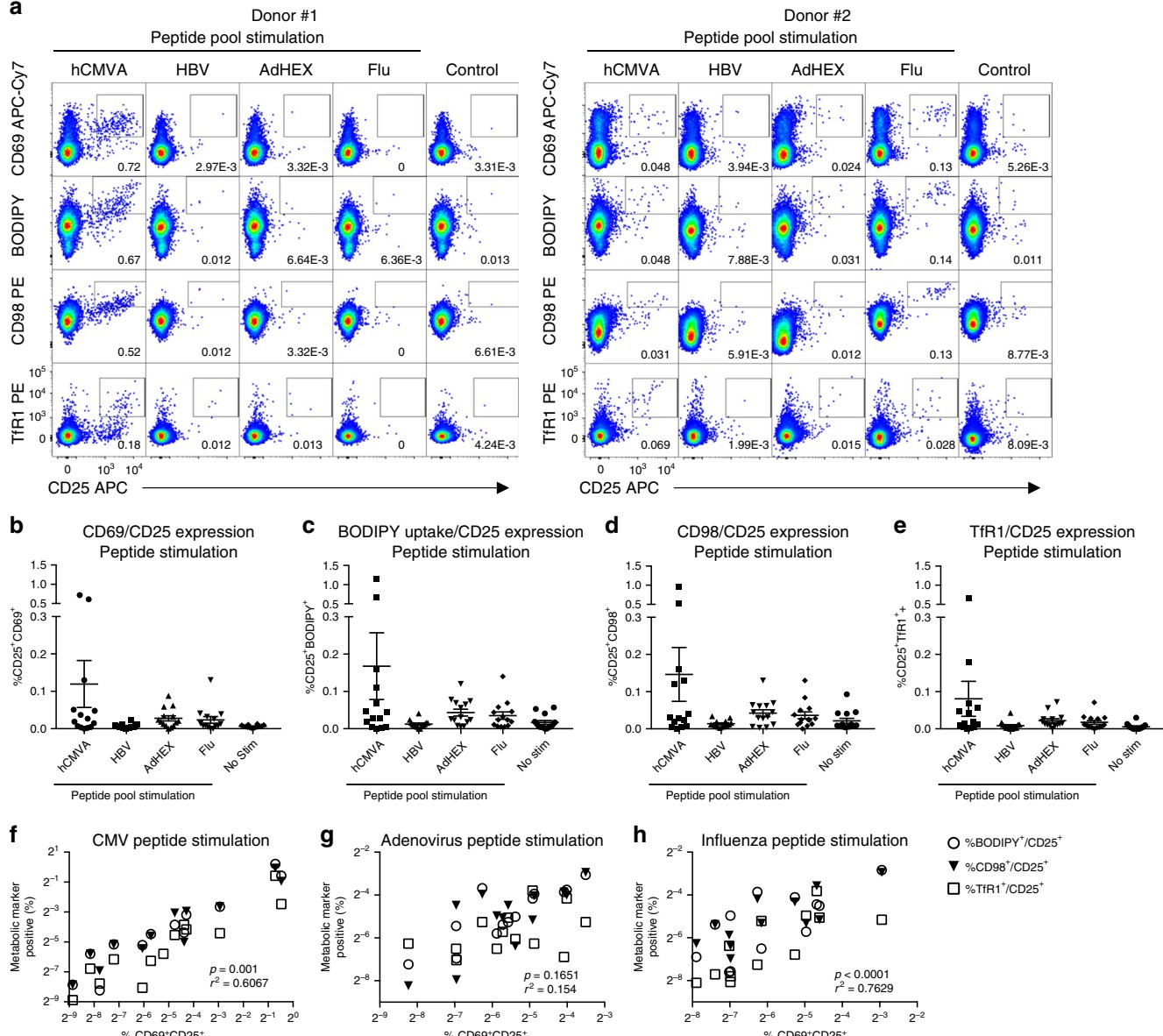

**Fig. 5** Antigen-specific in vitro CD8$^+$ T cell activation results in increased metabolite transporter expression and substrate utilization. **a** CD8$^+$ T cells from healthy donors were analyzed by flow cytometry after 48 h in vitro stimulation with 1 µg ml$^{-1}$ of the indicated peptide pool. T cell activation was assessed based on **b** upregulation of CD69 and CD25 expression, **c** increased uptake of BODIPY FL-C$_{16}$ and increased CD25 expression, **d** increased expression of CD98 and CD25, and **e** increased expression of TfR1 and CD25. Error bars show mean and SEM. The percentage of CD8 T cells exhibiting a metabolically activated state as assessed by BODIPY FL-C16 uptake, CD98 expression, and TfR1 expression following in vitro stimulation with **f** hCMV **g** adenovirus, or **h** influenza derived peptide stimulation was highly correlated with the percentage of cells expressing the conventional activation markers CD69 and CD25 at the same time point. Results are representative of two independent experiments with a total of 15 individual donors. Source data are provided as a Source Data file

on in vivo activated CD8$^+$ T cells can be used to determine their effector/memory potential, we utilized scRNAseq to assess the relative abundance of memory precursor CD8$^+$ T cell clonotypes within either the TfR1$^+$HLA-DR$^+$ or CD38$^+$HLA-DR$^+$ CD8$^+$ T cell compartment 14 days post-TAK-003 administration. To this end, TfR1$^+$HLA-DR$^+$ CD8$^+$ T cells from the same individual used in the previous scRNAseq analysis were sorted 14 days post-TAK-003 inoculation and subjected to scRNAseq analysis (Fig. 7a). This analysis resulted in the capture of 117 individual TAK-003-reactive cells with 80 unique TCR clonotypes (Fig. 7b, Supplementary Table 3). NS1- and NS3-reactive memory precursors within the sorted TfR1$^+$HLA-DR$^+$ CD8$^+$ T cell

pool were identified by the presence of TCR clonotypes found in NS1- and NS3-stimulated memory CD8$^+$ T cells 120 days post-vaccination (Fig. 2b, d, Table 1). Of the 117 captured TfR1$^+$HLA-DR$^+$ CD8$^+$ cells within the scRNAseq dataset, 31 and 24 were NS1- or NS3-reactive memory precursors, respectively (Fig. 7b). These numbers represent 47% of all cells within the sorted population, ~four-fold higher than the frequency of the same TCR clonotypes within the sorted CD38$^+$HLA-DR$^+$ CD8$^+$ T cell pool previously analyzed (Fig. 2d, Fig. 7b).

To determine if the expression of TfR1 accurately delineates DENV-reactive memory precursors within the larger CD38$^+$HLA-DR$^+$ CD8$^+$ T cell pool (Fig. 3, Cluster 1), we

performed scRNAseq gene expression analysis on the sorted TfR1⁺HLA-DR⁺ CD8⁺ T cells analyzed above. The resultant single cell gene-expression dataset was merged with the previously generated gene expression analysis of sorted CD38⁺HLA-DR⁺ CD8⁺ T cells (Fig. 3), and the relationship and overlap between the two populations assessed (Fig. 7c). As predicted by the previous flow cytometry analysis, TfR1⁺HLA-DR⁺ CD8⁺ T cells form a distinct transcriptional cluster, overlapping with a subset of the larger CD38⁺HLA-DR⁺ CD8⁺ T cell pool (Fig. 7c, Supplementary Table 4). TfR1⁺HLA-DR⁺ CD8⁺ T cells transcriptionally co-localize with (and are enriched in) cells expressing either NS1- or NS3-reactive T cell receptors. Furthermore, TfR1⁺HLA-DR⁺ CD8⁺ T cells co-localize exclusively with those CD38⁺HLA-DR⁺ CD8⁺ T cells formerly falling into the previously defined Cluster 1 identified in Fig. 3a. These data demonstrate that the expression of TfR1 is a robust marker for the identification and isolation of vaccine-reactive CD8⁺ T cells enriched for effector/memory potential.

To extend the observation that TfR1 expression may better define vaccine-reactive CD8⁺ T cells with effector/memory potential, we further analyzed CD8⁺ T cells from an additional 12 individuals 14 days after immunization with TAK-003 by flow cytometry with the addition of intracellular markers of CD8⁺ T cell effector function, proliferation, and effector/memory potential. As expected, only a subset of CD38⁺HLA-DR⁺ CD8⁺ T cells express TfR1 14 days post-TAK-003 administration (Fig. 7d, Supplementary Fig. 10). Additionally, we observed that TfR1 expression within the CD38⁺HLA-DR⁺ CD8⁺T cell compartment positivity correlates with the presence of markers of cellular proliferation (Ki67) (Fig. 7e, f), cytolytic function (Granzyme B) (Fig. 7e, g), and effector/memory lineage commitment (EOMES) (Fig. 7e, h). Expression of these markers were significantly enriched in TfR1⁺CD38⁺HLA-DR⁺ CD8⁺ T cells relative to TfR1⁻CD38⁺HLA-DR⁺ CD8⁺ T cells, or CD38⁻HLA-DR⁻ CD8⁺ T cells. These data demonstrate that the surface expression of TfR1 is a marker of effector/memory potential, and may aid in the identification and characterization of vaccine-reactive T cells within the total activated CD8⁺ T cell pool (Supplementary Fig. 11).

## Discussion

In this study, we demonstrate that the live-attenuated tetravalent DENV vaccine TAK-003 is capable of eliciting potent and durable cellular immunity. CD8⁺ T cell activation in response to TAK-003 administration peaked 28 days post-vaccination, while maximal CD4⁺ T cell expansion occurred on day 14. DENV-specific cellular immunity persisted for at least 120 days following immunization as assessed by IFN-γ ELISPOT. The antigenic specificity of the cellular memory immune response elicited by TAK-003 spanned the entire DENV proteome and exhibited significant cross-reactivity against all four DENV serotypes. Analysis of the clonotypic and functional diversity of TAK-003-stimulated CD8⁺ T cells 14 days after vaccination utilizing scRNAseq revealed a significant amount of transcriptional heterogeneity within a phenotypically homogenous population. Isolation and scRNAseq-based analysis demonstrated that the dominant TCR clones within the NS1- and NS3-reactive memory CD8⁺ T cell populations assessed 120 days post-vaccination can also be observed within the activated CD38⁺HLA-DR⁺CD8⁺ T cell compartment 14 days post-vaccination. scRNAseq-based analysis of these memory precursor cells present at day 14 post-vaccination revealed a unique transcriptional signature, dominated by the gene expression pathways associated with cellular metabolism and proliferation.

Based on these observations, we were able to develop a panel of markers to assess the metabolic potential of both CD8⁺ and CD4⁺ T cells following in vitro or in vivo activation. We were able to demonstrate that surface expression of TfR1 (transferrin receptor) marks cells with the highest functional and proliferative capacity, as assessed by Ki67, Granzyme B, and EOMES expression, providing a robust marker for identifying the CD8⁺ T cells shortly after vaccination with the greatest effector/memory potential. These data not only provide insight into molecular mechanisms responsible for regulating memory T cell development, but also suggest possible therapeutic targets for enhancing vaccine efficacy by selectively priming the metabolism of effector/memory precursor CD8⁺ T cells during the critical para-vaccine T cell expansion phase. We believe these data demonstrate that highly activated effector cells and memory precursor cells are not categorically incompatible descriptors, as a significant fraction of the memory CD8⁺ T cells generated in response to TAK-003 vaccination clonally overlap with a population of cells with a proliferative and highly activated transcriptional profile present shortly after vaccination.

The regulation of memory T cell development and homeostasis is a complex and incompletely understood process, involving the integration of a constellation of immunological cues such as antigen density[44], TCR/peptide/MHC affinity[45], duration of antigen exposure[46], and cytokine availability[47–50]. However, it is becoming increasingly clear that the development of a stable memory T cell population is dependent on the availability of a handful of key metabolites and the expression of a corresponding metabolic cellular program[39,51,52]. In particular, the availability of glucose[53,54], long-chain fatty acids[42,55], amino acids[56–58], and micronutrients such as iron[59,60] can profoundly impact CD8⁺ T cell effector and/or memory potential. Depriving T cells of any of these key metabolites either through pharmacological or genetic means has significant implications for T cell development, effector function, and long-term persistence.

Directly manipulating the metabolism of T cells in vitro or in vivo to influence effector function or persistence has primarily focused on restricting or enhancing access to the metabolite glucose. Due to the unique metabolic requirements of nascently activated T cells, which overwhelmingly eschew conventional mitochondrial oxidative phosphorylation in favor of oxidative glycolysis[39,51,52], glucose metabolism is a convenient therapeutic target. The proliferation and terminal effector function of both CD4⁺ and CD8⁺ T cells can be significantly enhanced by increasing glucose bioavailability[54,61], whereas restricting glucose metabolism can facilitate the development of long-lived and/or suppressive T cell lineages[53,62]. However, the utility of manipulating systemic glucose metabolism for therapeutic immunoregulatory effect in vivo is questionable. Systemic glucose metabolism is tightly regulated, and even modest perturbations in systemic glucose availability can have profound negative consequences for the well-being of patients. However, the selective upregulation of TfR1 on vaccine-reactive CD8⁺ T cells suggests that manipulating iron availability following vaccination may selectively enhance the expansion of the most functional vaccine-reactive CD8⁺ T cells. Indeed, loss-of-function mutations of TfR1 have been observed and are associated with severe defects in adaptive immune function[63]. Furthermore, expression of the gene products highlighted in this study—such as TfR1—is not restricted to vaccine-elicited T cell activation, as natural infection with dengue is also associated with increased TfR1 expression on activated T cells[64].

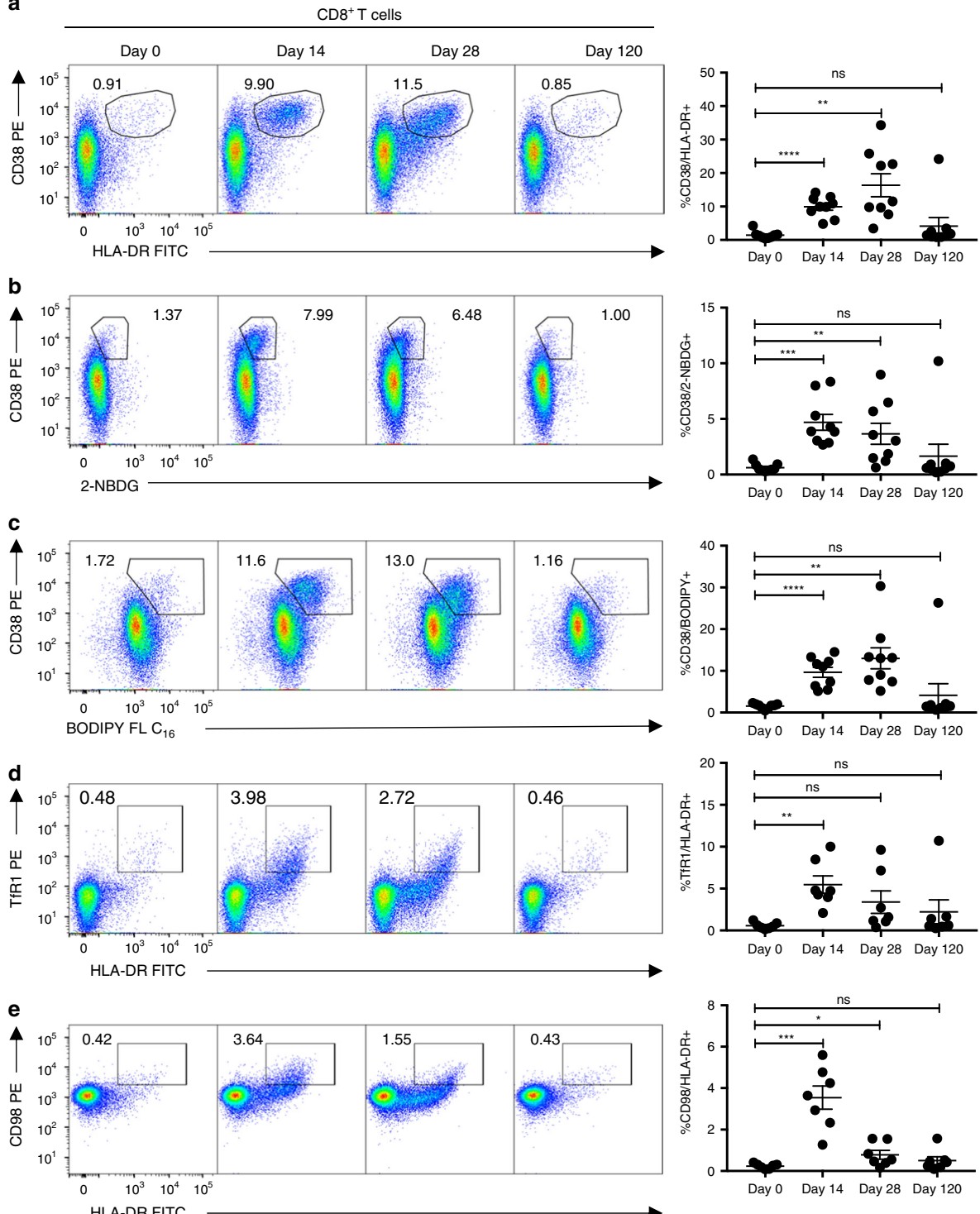

**Fig. 6** Vaccine-reactive T cells can be identified by changes in metabolite transporter expression and metabolite utilization. CD8+ T cells from TAK-003 recipients were analyzed by flow cytometry at days 0, 14, 28 and 120 post-vaccination. Vaccine-reactive CD8+ T cells were quantified based on expression of **a** CD38/HLA-DR, **b** CD38/2-NBDG, **c** CD38/BODIPY FL-C$_{16}$, **d** TfR1/HLA-DR, and **e** CD98/HLA-DR. Error bars show mean and SEM. $n = 10$ individuals. *$P < 0.05$, ***$P < 0.001$, ****$P < 0.0001$ (Paired two-tailed $t$-test). Source data are provided as a Source Data file

These findings suggest the preferential survival of T cells undergoing clonal expansion in vivo is dependent on metabolite availability and the initiation of a transcriptional program permissive to nutrient uptake. Cumulatively, these data highlight the utility of high-content single-cell transcriptomic analysis coupled with more traditional cellular immune monitoring in assessing vaccine-elicited T cell immunity. The ability to accurately and longitudinally track T cell clones from acute infection to stable memory provides a unique opportunity to identify correlates of T cell-mediated immunity with single-cell resolution. Future work is required to verify the distinctiveness of the metabolic and transcriptional programming that determines long-term T

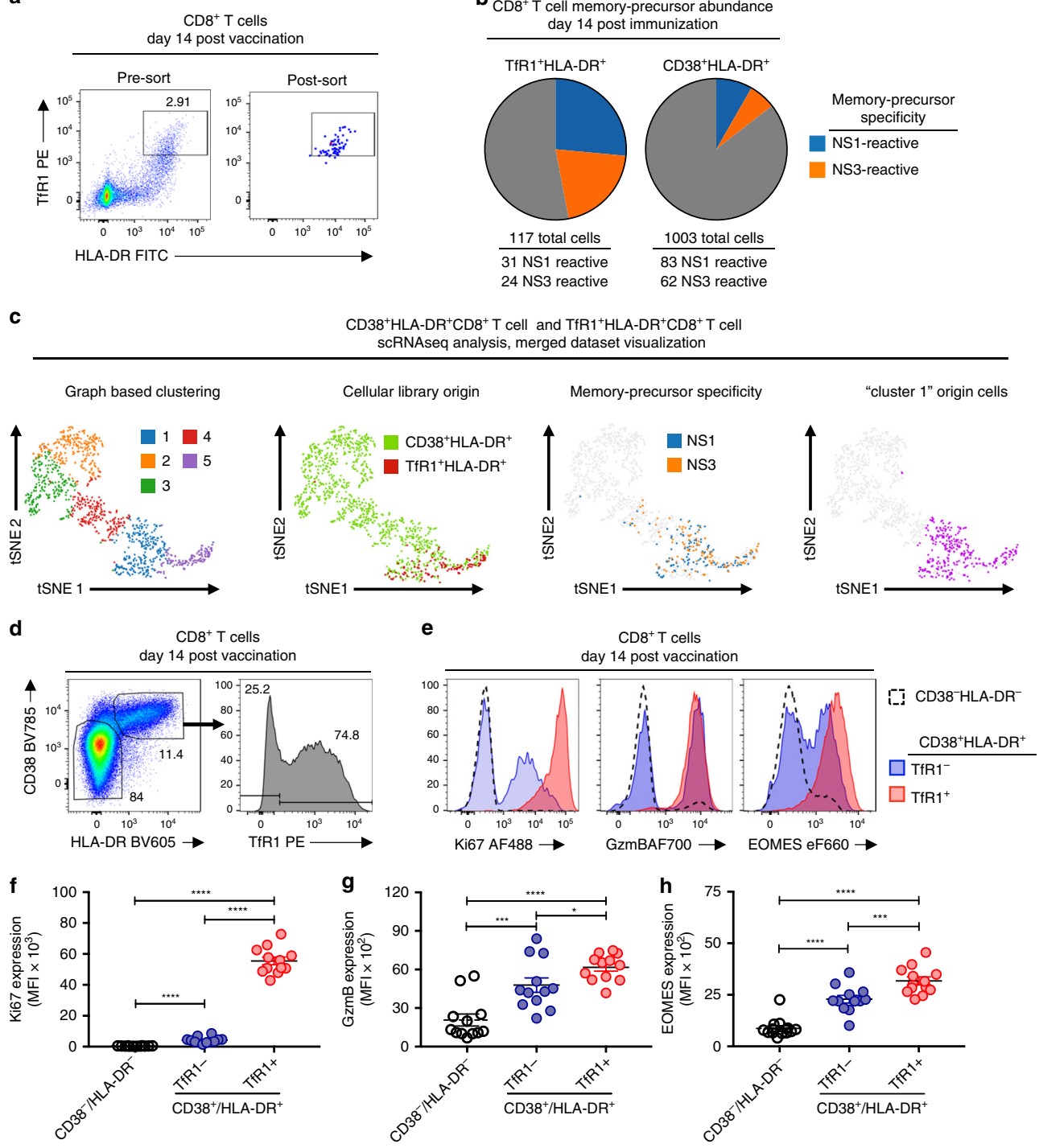

**Fig. 7** TfR1 expression correlates with CD8[+] T cell effector/memory potential in vaccine-reactive CD8[+] T cells. CD8[+] T cells from TAK-003-inoculated individuals were analyzed 14 days after immunization. **a** TfR1[+]HLA-DR[+] CD8[+] T cells were sorted from a TAK-003 inoculated individual 14 days post immunization and subjected to single-cell RNAseq analysis. **b** The abundance of memory-precursor clonotypes was assessed at day 14 post-vaccination within the TfR1[+]HLA-DR[+] or CD38[+]HLA-DR[+] CD8[+] T cell compartments. Memory-precursor clonotypes were defined as TCR clones found at both day 14 within the pools of phenotypically activated CD8[+] T cells, and at day 120 within either NS1 or NS3 reactive memory CD8[+] T cell populations. **c** scRNAseq gene expression analysis of sorted CD38[+]HLA-DR[+] CD8[+] T cells and TfR1[+]HLA-DR[+] CD8[+] T cells 14 days post TAK-003 administration. Merged datasets showing cellular library origin, memory precursor specificity, and Cluster 1 origin cells. **d** Flow cytometric analysis of TAK-003-reactive CD8[+] T cells 14 days post-vaccination. Cells were subdivided into CD38[−]HLA-DR[−], and CD38[+]HLA-DR[+] TfR1[−], and CD38[+]HLA-DR[+] TfR1[+] populations, then assessed for intracellular expression of **e** Ki67, GzmB, and EOMES. CD71 expression on CD8[+] T cells corresponds with significantly higher intracellular levels of **f** Ki67, **g** GzmB, and **h** EOMES. Error bars show mean and SEM. $n = 12$ individuals. *$P < 0.05$, ***$P < 0.001$, ****$P < 0.0001$ (Paired two-tailed $t$-test). Source data are provided as a Source Data file

memory and whether these markers define the protective capacity of T cell immunity.

## Methods

**Cells/samples**. The samples used in this study were collected during a Phase 1 trial in US adults of a tetravalent, live-attenuated dengue virus vaccine candidate, TAK-003 (NCT01728792; WRAIR #1987), as well as from healthy US adult volunteers (WRAIR #1868). Whole blood was collected in Cell Preparation Tubes (BD Vacutainer) for isolation of PBMC. Cells were cryopreserved at ~$10^7$ per mL and stored in vapor-phase liquid nitrogen until use. Vaccine administration and PBMC collection were performed after written informed consent. The studies were approved by the institutional review boards at the State University of New York Upstate Medical University and the Human Subjects Research Review Board for the Commanding General of the U.S. Army Medical Research and Material Command.

**T cell ELISpot assay**. Cryopreserved PBMC were thawed and placed in RPMI 1640 medium supplemented with 10% heat-inactivated normal human serum (100–318, Gemini Bio-Products), L-glutamine, penicillin, and streptomycin. After an overnight rest at 37 °C, the PBMC were washed, resuspended in serum-free medium (SFM; X-VIVO 15, Lonza), and $1–2 × 10^5$ cells were plated per well of a 96-well Millipore MAIPSWU plate coated with anti-IFNγ antibody according to the manufacturer's instructions (3420-2HW-Plus, Mabtech Inc.). Peptide pools were added to the cells at a final concentration of 1 μg/mL/peptide prior to incubation at 37 °C overnight. Controls included SFM plus 0.5% DMSO (negative) and anti-CD3 (positive). The ELISpot plates were developed using TMB substrate and read using a CTL-ImmunoSpot® S6 Ultimate-V Analyzer (Cellular Technology Limited). For the T cell epitope mapping studies using matrixed peptide pools, positive wells were defined as those with SFC/$10^6$ PBMC values five-fold over background (negative/no stimulation control) and greater than 5 SFC/$10^6$ PBMC after subtraction of the negative control.

**Peptides**. Overlapping peptide pools corresponding to the full-length envelope (E), non-structural 1 (NS1), NS2, NS3, NS4, and NS5 proteins for DENV-1-4 were obtained through the NIH Biodefense and Emerging Infections Research Resources Repository, NIAID, NIH (Supplementary Table 5). Additional overlapping peptide pools covering the capsid (C) and precursor membrane (prM) proteins of DENV-1-4, pp65 protein from hCMV, nucleocapsid protein from influenza H3N2, hexon protein from adenovirus serotype 3, and Large Envelope protein from HBV were purchased from JPT Peptide Technologies (Supplementary Table 5). Peptide pool stocks were reconstituted in DMSO at a concentration of 200μg/mL/peptide and stored at −80 °C.

**In vitro T cell stimulation**. Polyclonal T cell activation was performed by stimulating healthy donor PBMCs at a concentration of $5 × 10^6$ cells/mL with 0.1 μg/mL αCD3 (Clone OKT3, Biolegend 317315) and 1 μg/mL αCD28 (Clone CD28.2, BD 555725) in complete cell culture media consisting of RPMI 1640 supplemented with 10% heat-inactivated fetal bovine serum (Sigma), 2 mM L-glutamine (Lonza/BioWhittaker), and 100 U/mL penicillin/streptomycin (Lonza/BioWhittaker). For antigen-specific T cell simulation, healthy donor PBMCs were resuspended at a concentration of $5 × 10^6$ cells/mL in complete cell culture medium and stimulated with the indicated peptide pool at a final concentration of 1 μg/mL. Assessment of antigen-specific cytokine production was performed by resting PMBC overnight in complete cell culture medium, followed by a 6-hour stimulation with the indicated peptide pool in the presence of GolgiStop (BD Biosciences, 554724) and GolgiPlug (BD Biosciences, 55029), each at a 1:1000 dilution.

**Flow cytometry**. Surface staining for flow cytometry analysis was performed in PBS supplemented with 2% FBS at room temperature. Aqua Live/Dead (ThermoFisher, L34957) was used to exclude dead cells in all experiments. Intracellular protein staining was performed using the Foxp3 Fixation/Permeabilization kit (ThermoFisher, 00-5523-00) according to the manufacturer's recommendation. For analysis of intracellular cytokine production following in vitro peptide stimulation, cells were fixed with 4% PFA in PBS and permeabilized with 1X Perm/Wash buffer (BD Bioscience, 554723). Antibodies and dilutions used for flow cytometry analysis are listed in Supplementary Fig. 6. Flow cytometry analysis was performed on a custom-order BD LSRFortessa instrument and analyzed using FlowJo v10.2 software (Treestar). Cell sorting was performed on a BD FACSAria Fusion instrument.

**Isolation of DENV-reactive memory CD8+ T cells**. NS1- and NS3-reactive memory CD8+ T cells were identified and isolated from PBMC samples obtained 120 days post-vaccination. Cryopreserved PBMC samples were thawed and resuspended in complete cell culture media at a concentration of $5 × 10^6$ cell/mL and stimulated with 1 μg/mL of NS1- or NS3-derived peptide pools (Supplementary Table 5) for 18 h at 37 °C. NS1- or NS3-reactive CD8+ T cells were identified by expression of the activation markers CD25 and CD69, and isolated by flow sorting.

**Metabolite uptake assay**. Cryopreserved PBMC samples were thawed and resuspended in complete cell culture media at a concentration of $5 × 10^6$ cell/mL and rested for 30 min at 37 °C prior to metabolic analysis. For assessment of glucose uptake, cells were subsequently washed 2X with glucose-free RPMI (ThermoFisher, 11879020), resuspended at a concentration of $5 × 10^6$ cell/mL in glucose-free RPMI, and rested for an additional 10 min at 37 °C. 2-NBDG (ThermoFisher, N13195) was added to a final concentration of 100 μM, and cells were incubated for 30 min at 37 °C. Cells were washed 2X with PBS + 5% FBS, then surface stained for flow cytometry analysis as described above. For assessment of fatty-acid uptake, BODIPY FL-$C_{16}$ (ThermoFisher, D3821) was added to cells in complete cell culture media at a final concentration of 1 μM. Cells were incubated for 30 min at 37 °C, then washed 2X with PBS + 5% FBS and surfaced stained for flow cytometry analysis as described above. For assessment of transferrin uptake, transferrin-AF488 (ThermoFisher, T13342) was added to cells in serum-free RPMI at a final concentration of 50 μg/mL. Cells were incubated for 30 min at 37 °C, then washed twice with PBS + 5% FBS and surfaced stained for flow cytometry analysis as described above. Binding specificity of 2-NBDG and BODIPY FL-$C_{16}$ on activated T cells was assessed by titration of D-glucose (Sigma, G8270) or Oleic Acid-Albumin (Sigma, O4008), respectively.

**Single-cell RNA sequencing library generation**. Flow-sorted CD8+ T cell suspensions at a density of 50–500 cells/μL in PBS plus 0.5% FBS were prepared for single-cell RNA sequencing using the Chromium Single-Cell 5′ Reagent version 2 kit and Chromium Single-Cell 5′ Controller (10x Genomics, CA)[45]. In short, 500–2000 cells per reaction were loaded for gel bead-in-emulsion (GEM) generation and barcoding. Reverse transcription, RT-cleanup, and cDNA amplification were performed to isolate and amplify cDNA for downstream 5′ gene or enriched V(D)J library construction according to the manufacture's protocol. Libraries were constructed using the Chromium Single-Cell 5′ reagent kit, V(D)J Human T Cell Enrichment Kit, 3′/5′ Library Construction Kit, and i7 Multiplex Kit (10x Genomics, CA) according to the manufacturer's protocol.

**Sequencing**. scRNAseq 5′ gene expression libraries were sequenced on an Illumina NextSeq platform with a 500/550 High Output Kit v2 (150 cycles) to a read depth of ~30,000 reads/cell. Sequencing parameters were set for Read1 (26 cycles), Index1 (eight cycles), and Read2 (98 cycles). scRNAseq TCR V(D)J enriched library sequencing was performed on an Illumina MiSeq platform with a v3 Reagent Kit (600 cycles) to a read depth of ~10,000 reads/cell. Sequencing cycles were set at 150 for Read1, 8 for Index1), and 150 for Read2. Prior to sequencing, library quality and concentration were assessed using an Agilent 4200 TapeStation with High Sensitivity D5000 ScreenTape Assay and Qubit Fluorometer (Thermo Fisher Scientific) with dsDNA BR assay kit according to the manufacturer's recommendation.

**TCR sequence analysis**. Sorted CD8+ T cell TCR clonotype identification, alignment, and annotation was performed using the 10x Genomics Cell Ranger pipeline. Sample demultiplexing and clonotype alignment was performed using the Cell Ranger software package (10x Genomics, CA, v2.1.0) according to the manufacturer's recommendations, with the default settings, and mkfastq/vdj commands, respectively. TCR clonotype alignment was performed against a filtered human V(D)J reference library generated using the Cell Ranger mkvdjref command and the Ensembl GRCh38 v87 top-level genome FASTA and the corresponding Ensembl v87 gene GTF. TCR clonotype visualization, diversity assessment, and analysis were performed using the Loupe VDJ Browser (10x Genomics, CA, v2.0.0). TCR gene segment usage was assessed and visualized using VDJTools[65]. TCR clonal overlap between NS1- and NS3-specific memory CD8 T cells and putative memory precursors was assessed by using the CDR3nt sequence of defined memory cells and the corresponding full-length clonotype as the search seed. T cells isolated at day 14 which shared the same CDR3nt sequence and corresponding clonotype as a defined memory cell were defined as memory precursors. Cells with only a full-length alpha-or-beta chain were matched based on only the overlap of the single chain.

**10x Genomics 5′ gene-expression data analysis**. 5′ gene expression analysis from sorted CD8+ T cells was performed using the 10x Genomics Cell Ranger pipeline[45]. In short, sample demultiplexing and analysis was performed using the Cell Ranger software package (10x Genomics, CA, v2.1.0) according to the manufacturer's recommendations, with the default settings and mkfastq/count commands, respectively. Transcript alignment was performed against a human reference library generated using the Cell Ranger mkref command and the Ensembl GRCh38 v87 top-level genome FASTA and the corresponding Ensembl v87 gene GTF. Data visualization and differential gene expression analysis were performed using the Loupe Cell Browser (10x Genomics, CA, v2.0.0). t-SNE plot visualization of gene expression data was based on the cellular coordinates calculated by the Cell Ranger count command. Cell Ranger count outputs were subsequently filtered to only contain cells with a recovered TCR sequence as identified above, and to contain cells not expressing CD14 (monocytes) or CD19 (B cells) using the Cell Ranger reanalyze command. Aggregation and re-analysis of multiple gene expression datasets was performed using the Cell Ranger aggr command. Read-

depth normalization was performed on all merged datasets by subsampling mapped reads to achieve an equal number of confidently mapped reads per cell. Differential gene expression in Cell Ranger was calculated using a negative binomial exact test in sSeq[66], paired with a fast asymptotic beta test in edgeR[67,68] for samples with a large number of counts. For each unique cluster, the algorithm was run on that cluster relative to all other clusters, generating a list of genes that were differentially expressed in the cluster of interest relative to all other cells in the sample. The top 20 differentially expressed genes in each cluster are listed in the relevant tables, first ranked by p-value (using the Benhamini-Hochberg procedure to control for FDR), then ranked by fold-change. Differentially expressed genes that failed to reach a Benhamini-Hochberg adjusted *p*-value of 0.05 were still listed, but marked as non-significant in the relevant tables.

**Statistical analysis**. All statistical analysis was performed using GraphPad Prism 6 Software (GraphPad Software, La Jolla, CA). A *P*-value < 0.05 was considered significant.

**Reporting summary**. Further information on research design is available in the Nature Research Reporting Summary linked to this article.

## Data availability

The authors declare that all data supporting the findings of this study are available within this article and its Supplementary Information files, or from the corresponding author upon reasonable request. Single-cell RNAseq gene expression data have been deposited in the Gene Expression Omnibus database under the accession code GSE132950. The source data underlying Figs. 1b, d, f, 4, 5 b, c, d, e, f, g, h, 6a, b, c, d, e, 7f, g, h, 1c, d, 6, 7a, b, 9a, b, c, d, e are provided as a Source Data File.

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

## Acknowledgements

This work was supported by the Military Infectious Disease Research Program (MIDRP) and the Congressionally Directed Medical Research Program (CDMRP). This research was performed while Dr. Waickman held an NRC Research Associate award at the Walter Reed Army Institute of Research. CLM was supported by NIH COBRE grant P20 GM104317.

## Author contributions

A.T.W. conceived of the project, designed, and executed experiments, analyzed data, and wrote the paper.. K.V., T.L., and K.H. generated data. W.R. and H.F. designed and executed experiments and analyzed data. C.M. and B.G. analyzed data. R.G.J. and J.R.C. provided project oversite and secured funding.

## Additional information

**Competing interests:** The authors declare no competing interests.

**Disclaimer:** The opinions or assertions contained herein are the private views of the authors and are not to be construed as reflecting the official views of the US Army or the US Department of Defense. Material has been reviewed by the Walter Reed Army Institute of Research. There is no objection to its presentation and/or publication. The investigators have adhered to the policies for protection of human subjects as prescribed in AR 70–25.

