## [Peer Review File · Nature Communications]

Reviewers' Comments:

Reviewer #1:

Remarks to the Author:

The manuscript shows novel data on the genomics and phenotype of T cells following vaccination with DENV Vaccine from Takeda, which is a tetravalent vaccine that generates T cell responses.

The main message of the paper is that T cell induced by this vaccine undergo metabolic programming since very early phase post-vaccination, and that memory precursor exist with a specific signature that give rise to long term memory phenotype.

Overall, I found the results of the paper very interesting and worth publishing.

However, the story sometimes lack focus, and some of the experimental data, especially scRNAseq, could be presented more thoroughly.

Comments

- The single cell RNAseq experiments have been designed in a way that I find hard to follow. Day 14 is about active T cells CD38+ HLA DR+, compared to double negative. But the scRNAseq for the control has not been used.
- For day 120, as memory cells have been first stimulated it is very difficult to interpret scRNAseq data (which have not been shown, only TCR). Did the authors consider the analysis of these cells without stimulation, maybe with tetramers?
- Can the authors explain whether Day 14 TCR repertoire from CD38+ and CD38- subset overlap?
- It is unclear whether scRNAseq have evidence of CD71 expression
- At day 14 scRNAseq, I would suggest a more detailed analysis of the gene expression data, Not only testing the expression level of CD71 and other markers that have been associated to metabolic activities, but also to investigate whether or not specific subset of T cells exists. One way would be to test the presence of memory precursor versus effector cells,
- In general, I would carefully analysis the scRNAseq data and not simply use cellRanger. There is a lot out there, in terms of software tools that have been shown to be more effective in detecting signals, and also to analysis drop out., Is CD71 affected by drop out?
- Also, I was expecting some analysis that compare scRNAseq data from Day 14 and later time point. Interesting the later time point has been analysis only for TCR, what about the genes?
- Why the authors have analysed only CD38+ cells in the gene expression analysis? Why not the CD38- population? What gene signatures these have and how they differ from the CD38 +? Did the metabolic pathways differ? Again, the gene expression analysis is very poor, with only a clustering analysis.
- It would be interesting to look at the evolutionary dynamics by comparing scRNAseq at day 14 and day 120. A pseudo time analysis would be probably helpful.
- Regarding peptide stimulation and ELISPOT, would be helpful to clarify whether specific peptides could be tested for each region and have a map against serotype?
- On this matter, the data show NS3 and NS1 reactive memory but not NS5, which is also a known highly immunogenic region of the virus. ELISPOT in Fig 2 show some response against NS5 peptides, Fig 2 This pie chart is only for one subject (Subject 30), Are these data on other subjects, consistent with these? Perhaps other subjects have a broader and stronger NS5 restricted response?
- On the message "". Interesting conclusion. Have the authors considered to test for specific property of long-term memory? Vaccine studies have been recently showing the presence of these precursor memory, see Rafi Ahmed paper in Nature on YFV and memory stem cells. This approach will allow investigation of activation but not memory.
- The authors should clarify the message of this work. I found the discovery of a subset of precursor

memory that could potentially be present already at day 14 the most interesting finding. Can the author retrospectively analyse the earliest time points?

- Is there data before day 14?
- The introduction of the work on CD71 CD98 comes a bit out of the blue. I see the point that gene expression suggests metabolic changes between the populations of T cells, But why these two markers? Gene expression didn't suggest these but an overall metabolic evolution. There are many other ways to investigate metabolic activities and also different pathways to be investigated, for instance, gene expression seems to suggest overexpression of MTOR and OHPHOS in some clusters (Cluster 1). Then point I am making is that this approach was not driven by the data but decided a priori as a hypothesis. This may not reflect the heterogeneity observed from the single cell data. Also, unclear whether this is the right mechanism that distinguish memory precursor from the rest of the memory cells.
- For Figure 4, why the authors have not performed the experiment on specific memory and effector subsets rather than total CD4 or CD8?

Minor point

I would specify in the abstract what metabolic programs determine T cell generation during early phase.

Reviewer #2:

Remarks to the Author:

I have looked at the paper and realise that part of it is outside my area of expertise. I am not familiar with single cell RNA sequencing data so cannot comment on that. I am also not familiar with work looking at human T cell responses to vaccines so that I do hope you have got an 'expert vaccine immunologist looking at this paper.

In terms of importance of the subject matter then I do feel that understanding the responses triggered by vaccines is well worth analysing and hence this paper addresses an important area of immunology and science.

The part of the work I am familiar with is the metabolic profiling assays. Here the authors use a number of tools to examine how a Dengue virus stimulates T cell metabolism. They use CD71 and CD98 antibodies which detect respectively the transferrin receptor and the heavy subunit of system L amino acid transporters. These are well validated tools and the data here should be robust.

The in vitro validation of these markers was good to see but not particularly novel. However it was very interesting to see that one could use CD71 to 'sense' a good vaccine response in T cells isolated from immunised individuals ex vivo . It was also interesting to see that it could be used as an assay to monitor T cell activation in for ex vivo T cells stimulated in vitro with various antigen combinations. Here I was surprised that this had not been done before. CD71 and CD98 are common markers for activated T cells that have been in existence since the 1980s? They are known to identify transformed and activated lymphocytes. None the less this is interesting to see these responses although I was not really convinced that CD71 and CD98 were any more sensitive than some of the other activation markers.

The authors also use two fluorescent dyes 2NPDG and BODIPY FL-C in credible attempts to monitor glucose uptake and fatty acid uptake. The problem with these experiments is that these tool compounds need to be used very carefully with carefully conceived controls to avoid problems with

non specific binding to large cells versus small cells. These controls are not used in the paper or discussed. The controls are easy to do and simply require that the authors show that the binding of NPDG can be competed with excess glucose. ie cold competition of the glucose transporters – other wise there is no way to tell if this uptake is via a transporter or not or what it measures . Similarly with the fatty acid dye- Can a non fluorescent analogue compete? The authors conclude that the data they show reflect changes in expression of fatty acid transporters and glucose transporters – it does not. The assay they use is simply not robust enough. The controls need to be done before these data can be cited to reflect transport.

However, given that the best correlation is with iron transport then the CD71 data is perhaps enough? A simple story based on CD71 is interesting given that we know how important iron metabolism is for cells.

Here other assays the authors might like to consider in future in terms of in vitro re-challenge would be to assess if the uptake of fluorescent transferrin gave more sensitivity? As well, single cell assays for ERK1/2 phosphorylation and S6 phosphorylation can be very useful readouts of T cell activation in vitro and extremely sensitive. Moreover, antigen receptor engagement activates Erks1/2 with a couple of minutes and even S6 phosphorylation takes only 30 minutes. No need to do the long time point restimulation.

Reviewer #3:

Remarks to the Author:

The major claim from this study is that a unique transcriptional signature in a subset of the T-cell effectors expanding in response to dengue TAK-003 vaccination identifies them as memory precursors. CD71 is a major phenotypic marker for this subset.

This reviewer appreciates the excellent quality of the work, the extensive characterization of the T-cell expansion in dengue vaccine recipients using RNA-seq, clonotypic analysis, and flow cytometry methods. The resultant information that will be of great use to the field.

However, the conclusion that CD71 (or other metabolic signatures) identifies the memory precursors, although has merit, is not convincing enough, with the data in its present form. The reasons are the following:

1. The authors sorted memory cells based on their functional specificity (i.e., ability to respond to NS1 or NS3 peptides by up-regulating CD69 and CD25), identified TCR clonotypes and examined the representation of these similar clonotypes in effector cells. However, the effector cells were sorted solely based on the expression of activation markers (CD38, HA-DR and or CD71) rather than functional specificity. In the absence of the information on specificity of these effector cells, it is difficult for this reviewer to be convinced of this major conclusion.
2. Many studies, including this study, suggests that CD71 is robustly expressed on proliferating cells. It is conceivable that these proliferating cells could have been highly enriched for non-structural protein specific cells, potentially due to higher abundance of these NS protein derived peptides; and as a result, could give an impression that these are memory precursors.

Reviewer #1:

The single cell RNAseq experiments have been designed in a way that I find hard to follow. Day 14 is about active T cells CD38+ HLA DR+, compared to double negative. But the scRNAseq for the control has not been used.

We agree that these data are informative and would have been useful to include. These data have been added to the revised manuscript as **Supplemental Figure 4** and **Supplemental Table 2**.

For day 120, as memory cells have been first stimulated it is very difficult to interpret scRNAseq data (which have not been shown, only TCR). Did the authors consider the analysis of these cells without stimulation, maybe with tetramers?

We agree that the isolation and analysis of un-manipulated DENV-reactive memory T cells would have been ideal for our analysis. We also agree that the transcriptional analysis of the *ex vivo* stimulated T cells is difficult to interpret and impossible to directly compare to un-manipulated samples from other time points.

The fundamental goal of this study was to identify and characterize putative memory-precursor CD8⁺ T cells shortly after vaccination, which meant that we wanted to capture the TCR sequences of as many antigen-specific memory CD8⁺ T cells as possible from the day 120 post vaccination time point. Tetramer staining and flow-sorting would allow for the isolation of un-manipulated memory T cells, but requires *a priori* knowledge of all the HLA/peptide complexes recognized by the memory T cell pool, and would only facilitate the isolation of those T cells binding those discrete antigens (as opposed to isolation of T cells with functional capacity). In contrast, stimulation of the memory T cells with overlapping peptide pools and flow-sorting cells responding to stimulation – as we did in this study – maximizes the number and diversity of T cell clones captured in the analysis. Given the goals of the study, we felt that this was the most prudent assay.

Can the authors explain whether Day 14 TCR repertoire from CD38+ and CD38- subset overlap?

We observed minimal overlap in the Day 14 TCR repertoires from the sorted CD38⁺ and CD38⁻ populations on day 14 post vaccination. Of the 1,885 TCR clones identified between the two populations, only 16 clones were found in both populations. Of these 16 clones, 6 clones corresponded to the semi-invariant TRAV1-2/TRAJ33 arrangement associated with Mucosal-Associated Invariant T cells (MAITs), which happens to be a predominant population in this subject. We have added this information to the text of manuscript (lines 195-199).

It is unclear whether scRNASeq have evidence of CD71 expression

We observed little CD71 expression in scRNAseq gene expression dataset. Our rationale for selecting this markers for additional analysis was poorly explained in the initial manuscript, and we have added additional information explaining our decision making process.

In short, the metabolically-associated gene products identified by our scRNAseq analysis and preferentially-expressed in memory-precursor CD8⁺ T cells were uniformly intracellular in origin, and therefore incompatible with additional scRNAseq analysis. Therefore, we selected a panel of cell surface markers and traceable metabolites that we hypothesized would be preferentially expressed in - or taken up by - memory-precursor CD8⁺ T cells based on the signaling pathways preferentially expressed in these cells. The criteria used to select these markers were that **1)** the surface proteins were directly regulated by the gene pathways differentially expressed in our populations of interest and/or **2)** the metabolites associated with these surface markers were utilized by metabolic pathways differentially expressed in our cell populations of interest.

To this end, we selected the transferrin receptor complex (TfR1/CD71) and the Large-neutral Amino Acid Transporter 1 (LAT1/CD98) for additional analysis. The expression and surface-localization of both of these transporters is regulated by mTOR/p70S6K signaling (pathways preferentially expressed in our memory precursor cells). Furthermore, the metabolites they import – iron and amino acids, respectively – are critical co-factors for mitochondrial oxidative

phosphorylation or direct catabolic and anabolic substrates required in proliferating T cells, both of which can additionally directly impact mTOR/p70S6K signaling.

At day 14 scRNAseq, I would suggest a more detailed analysis of the gene expression data, Not only testing the expression level of CD71 and other markers that have been associated to metabolic activities, but also to investigate whether or not specific subset of T cells exists. One way would be to test the presence of memory precursor versus effector cells,

The CD38⁺HLA-DR⁺ CD8⁺ T cell population isolated for this primary scRNAseq analysis are already highly refined, and would have canonically been thought to represent a relatively homogeneous population. The fact that we observed such transcriptional heterogeneity within this population was highly surprising, and was the inspiration for the subsequent analysis for this study. In addition, while extensively-validated markers are available to identify naïve, memory, and effector CD8⁺ T cell subsets in humans, the identification of memory-precursor cells remains challenging (until this point). Markers have been identified in mice which appears to correlate with memory-precursor potential (IL-7R α /KLRG1, for example), but the utility of these markers in humans is unclear.

However, we have performed additional scRNAseq analysis on the sorted CD38⁺HLA-DR⁺ (resting) CD8⁺ T cells isolated in this study, and have highlighted the cellular diversity present in this less refined population. Of particular note, we were able to identify both conventional naïve/memory CD8⁺ T cells in this sample, as well as a large population of “non-conventional” CD8⁺ Mucosal-Associated Invariant T (MAIT) cells. These data can now be found in Supplemental Figure 4 and Supplemental Table 2 in the revised manuscript. We have also added additional analysis of the transcriptional profile of the sorted CD38⁺HLA-DR⁺CD8⁺ T cells in Figure 3.

In general, I would carefully analysis the scRNAseq data and not simply use cellRanger. There is a lot out there, in terms of software tools that have been shown to be more effective in detecting signals, and also to analysis drop out., Is CD71 affected by drop out?

CD71 expression does indeed appear to be affected by drop out in our dataset, as we find very few copies of the gene product in our analysis. However, to test if CD71 protein expression does in fact phenotypically distinguish the population of putative memory precursor cells that we have identified in our analysis (despite the poor TFRC transcript recovery in our dataset), we have performed additional scRNAseq gene expression analysis on sorted CD71⁺ HLA-DR⁺ CD8⁺ T cells 14 days post TAK-003 administration from the same individual analyzed in the rest of the study.

As can be seen in the new Figure 7C, the sorted CD71⁺ CD8⁺ T cells form a distinct transcriptional cluster when aggregated with the less-refined CD38⁺HLA-DR⁺ CD8⁺ T cells isolated from the same individual at the same time-point. Sorted CD71⁺ CD8⁺T cells are enriched in the transcriptionally-defined clusters containing NS1- and NS3-reactive TCR clones identified in day 120 memory CD8⁺ T cells. In addition, when the cells contained within the previously defined “cluster 1” (as identified in Figure 3A) are highlighted in the aggregated dataset, the sorted CD71⁺ CD8⁺ T cells are entirely co-localized within this previously defined population. Therefore, we believe that CD71 surface expression faithfully marks a population of cells enriched in effector/memory potential, despite the poor recovery of the TFRC gene product in our scRNAseq data. This observation highlight the utility of assessing not just the presence/absence of individual gene products in scRNAseq datasets, but the relative enrichment of signaling pathways associated with populations of interest.

Also, I was expecting some analysis that compare scRNaseq data from Day 14 and later time point. Interesting the later time point has been analysis only for TCR, what about the genes?

Due to the additional *in vitro* manipulation performed on the “memory” T cells isolated at day 120 (overnight peptide stimulation), we feel that direct comparison of these samples to un-manipulated samples obtained at day 14 post TAK-003 administration is not advisable.

Why the authors have analysed only CD38+ cells in the gene expression analysis? Why not the CD38- population?

What gene signatures these have and how they differ from the CD38 +? Did the metabolic pathways differ? Again, the gene expression analysis is very poor, with only a clustering analysis.

We agree that these data would have been useful; therefore, these data have been added as **Supplemental Figure 4** and **Supplemental Table 2** in the revised manuscript.

It would be interesting to look at the evolutionary dynamics by comparing scRNAseq at day 14 and day 120. A pseudo time analysis would be probably helpful.

We completely agree that a pseudo time analysis tracing the transcriptional evolution of DENV-reactive clones from day 14 to day 120 would be extremely helpful if both populations were isolated and analyzed in an identical manner. However, due to the additional *ex vivo* manipulation performed on the NS1/NS3 reactive CD8⁺ T cells from the day 120 sample (overnight peptide stimulation), this comparison is not possible with our current dataset.

Regarding peptide stimulation and ELISPOT, would be helpful to clarify whether specific peptides could be tested for each region and have a map against serotype?

We have performed additional ELISPOT analysis using matrixed peptide pools on samples obtained 120 days post TAK-003 vaccination to map the immunodominant NS1 and NS3 epitopes recognized in the individual analyzed in our study. These data – along with the location of the antigens within the DENV2 polyprotein – have been added as **Supplemental Table 1** in the revised manuscript.

On this matter, the data show NS3 and NS1 reactive memory but not NS5, which is also a known highly immunogenic region of the virus. ELISPOT in Fig 2 show some response against NS5 peptides, Fig 2 This pie chart is only for one subject (Subject 30), Are these data on other subjects, consistent with these? Perhaps other subjects have a broader and stronger NS5 restricted response?

We observed a very interesting pattern of NS1/NS3/NS5 responsiveness in individuals receiving this vaccine product. Many individuals exhibited a strong NS3 + NS5 response following vaccination, and some individuals exhibited a strong NS1 response and a moderate NS3 response (as seen in the individual analyzed in this study). However, few-if-any individuals exhibited a strong NS1 + NS5 response. We believe this is attributable to differences in HLA genotype, but have not performed the requisite HLA genotyping analysis required to conclusively make such a statement. A more detailed analysis of the immunogenic patterns generated by TAK-003 administration is the subject of another manuscript which is pending publication.

The authors should clarify the message of this work. I found the discovery of a subset of precursor memory that could potentially be present already at day 14 the most interesting finding. Can the author retrospectively analyse the earliest time points?

We completely agree with the reviewer that attempting to determine the earliest time at which these putative memory-precursors appear following vaccination would be very interesting. Unfortunately, day 14 post vaccination is the earliest time point post vaccination that PBMCs were collected in this study.

Is there data before day 14?

The earliest PBMC collection point after vaccination in this study was at day 14.

The introduction of the work on CD71 CD98 comes a bit out of the blue. I see the point that gene expression suggests metabolic changes between the populations of T cells, But why these two markers? Gene expression didn't suggest these but an overall metabolic evolution. There are many other ways to investigate metabolic activities and also different pathways to be investigated, for instance, gene expression seems to suggest overexpression of MTOR

and OHPHOS in some clusters (Cluster 1). Then point I am making is that this approach was not driven by the data but decided a priori as a hypothesis. This may not reflect the heterogeneity observed from the single cell data. Also, unclear whether this is the right mechanism that distinguish memory precursor from the rest of the memory cells.

We agree that our rationale for selecting CD71 and CD98 for additional analysis was poorly explained in the initial manuscript, and we have added additional text (lines 276 – 292) explaining our decision making process. The selection of these markers was driven by the transcriptional data generated in this study, but tempered by the fact that all of the metabolically-associated gene products identified by scRNAseq analysis and preferentially-expressed in memory-precursor CD8⁺ T cells were uniformly intracellular in origin. We are cognizant of the fact that many genes products suffer from drop-out in scRNAseq datasets, and decided to see if we could identify cell surface markers that were not directly represented in our datasets, but are involved in the same gene pathways that are preferentially regulated in memory-precursor CD8⁺ T cells.

The expression and surface localization of CD71 and CD98 are both directly regulated by the mTOR/p70S6Ke signaling pathways, and the metabolites they import are utilized for mitochondrial oxidative phosphorylation. For these reasons, we felt they were reasonable surrogate markers for identifying our population of interest.

To further confirm that the surface expression of CD71 on CD8⁺ T cells faithfully marks those cells that have the unique transcriptional profile we are ascribing to memory-precursors, we have performed additional single cell gene expression analysis on sorted CD71⁺ T cells. These data can now be found in **Figure 7C**.

For Figure 4, why the authors have not performed the experiment on specific memory and effector subsets rather than total CD4 or CD8?

As can be seen in **Figure 4** and **Supplemental Figure 6**, unfractionated bulk CD4⁺ and CD8⁺ T cells – containing both memory and naïve T cells - are metabolically quiescent in the absence of acute antigenic stimulation, expressing little CD71/CD98 and exhibiting little 2-NBDG/BODIPY FL-C₁₆ uptake. Our intent with the experiment in Figure 4 was to **1)** demonstrate that the metabolic parameters we selected based on our scRNAseq analysis were compatible with conventional flow cytometry analysis, **2)** to establish the kinetics of the metabolic changes associated with T cell activation, and **3)** to determine how these changes correlated with the expression of conventional T cell activation markers. Although naïve and memory CD8⁺ and CD4⁺ T cells were contained within the samples analyzed, these two populations become indistinguishable after *in vitro* stimulation, as both populations adopt an activated/effector phenotype.

Performing this type of granular kinetic analysis on *in vivo* stimulated CD4⁺ and CD8⁺ T cells following TAK-003 administration and tracking the metabolic changes associated with the transition from naïve -> effector -> memory would have been extremely interesting, but the PBMC collection schedule in this study did not allow for this level of analytical resolution.

Reviewer #2:

The authors also use two fluorescent dyes 2NPDG and BODIPY FL-C in credible attempts to monitor glucose uptake and fatty acid uptake. The problem with these experiments is that these tool compounds need to be used very carefully with carefully conceived controls to avoid problems with non specific binding to large cells versus small cells. These controls are not used in the paper or discussed. The controls are easy to do and simply require that the authors show that the binding of NPDG can be competed with excess glucose. ie cold competition of the glucose transporters – other wise there is no way to tell if this uptake is via a transporter or not or what it measures . Similarly with the fatty acid dye- Can a non fluorescent analogue' compete? The authors conclude that the data they show reflect changes in expression of fatty acid transporters and glucose transporters – it does not. The assay they use is simply not robust enough. The controls need to be done before these data can be cited to reflect transport.

We have performed the suggested competition assay using *in vitro* activated CD4⁺ and CD8⁺ T cells, which can now be found in the revised **Supplemental Figure 6A**. In addition, we have added additional references in the manuscript highlighting previously published examples of 2-NBDG and BODIPY FL-C₁₆ used in T cell analysis.

As a technical note, out-competing 2-NBDG with D-glucose proved to be slightly challenging. At the concentration of 2-NBDG we used on our analysis – which we empirically determined provided the most robust staining in activated T cells and is comparable to other published reports – we were only able to achieve a ~20% reduction in 2-NBDG uptake upon the addition of “cold” D-glucose before cell viability started to suffer. This may in part be due to the previously reported high affinity of glucose/NBD conjugates for the GLUT1 glucose receptor, which is significantly higher than that observed for unlabeled D-Glucose (PMID 19393014).

Here other assays the authors might like to consider in future in terms of *in vitro* re-challenge would be to assess if the uptake of fluorescent transferrin gave more sensitivity?

We had not previously considered using fluorescently labeled transferrin in the setting of T cell activation, but upon the reviewer’s recommendation we tested it in our *in vitro* T cell activation assay and were very gratified to see that transferrin-AF488 robustly bound *in vitro* activated CD4⁺ and CD8⁺ T cells (**Supplemental figure 6B**). In light of this very robust observation, we had hoped to perform additional analysis of samples from TAK-003 recipients to see if *in vivo* stimulated T cells also preferentially bound transferrin-AF488, but unfortunately additional samples were not available for this analysis.

As well, single cell assays for ERK1/2 phosphorylation and S6 phosphorylation can be very useful readouts of T cell activation *in vitro* and extremely sensitive. Moreover, antigen receptor engagement activates Erks1/2 with a couple of minutes and even S6 phosphorylation takes only 30 minutes. No need to do the long time point restimulation.

We completely agree with the reviewer that phosphorylation of ERK and ribosomal S6 are extremely robust markers of T cell activation. In fact, rS6 S235/236 phosphorylation is one of the most dramatic and reliable markers of T cell activation that we have tested. However, the extensive fixation and permeabilization required to detect these phosphoproteins is incompatible with the RNA isolation needed to recover accurate TCR sequence information, so we instead opted for the longer stimulation conditions required to cause upregulation of CD25 and CD69 on CD8⁺ T cells.

Reviewer #3:

The authors sorted memory cells based on their functional specificity (i.e., ability to respond to NS1 or NS3 peptides by up-regulating CD69 and CD25), identified TCR clonotypes and examined the representation of these similar clonotypes in effector cells. However, the effector cells were sorted solely based on the expression of activation markers (CD38, HA-DR and or CD71) rather than functional specificity. In the absence of the information on specificity of these effector cells, it is difficult for this reviewer to be convinced of this major conclusion.

We were cautious in ascribing any explicit antigenic specificity or functional capability to the population of circulating CD38⁺HLA-DR⁺ CD8⁺ cells found 14 days post TAK-003 vaccination, instead simply referring to them as “vaccine-elicited” in the manuscript. Attempts to dissect the antigen specificity of the CD8⁺ T cells circulating 14 days after vaccination were not possible, because although a large fraction of circulating CD8⁺ T cells at this time exhibited an activated phenotype, very few – if any – CD8⁺ T cells functionally responded to *ex vivo* DENV peptide stimulation, as now shown in the new **Figure 3E** and **Figure 3F**.

However, the claims made in our study do not depend on the total antigen specificity profile of the CD8⁺ T cells activated in response to TAK-003 administration at day 14. Rather, our data demonstrates that within this heterogeneous population of activated CD8⁺ T cells there is a discrete population of cells – for which the total antigen specificity is unclear – that share the same TCR sequences found in the DENV-reactive memory CD8⁺ T cell population 120 days post vaccination.

Many studies, including this study, suggests that CD71 is robustly expressed on proliferating cells. It is conceivable that these proliferating cells could have been highly enriched for non-structural protein specific cells, potentially due to higher abundance of these NS protein derived peptides; and as a result, could give an impression that these are memory precursors.

We completely agree with the reviewer that those cells which we have identified as being NS1- and NS3-reactive at day 14 post TAK-003 administration are likely to be highly proliferative due to their gene expression profile. However, it is crucial to note that the antigen specificity of cells isolated and analyzed 14 days post vaccination was assigned due to the shared expression of identical TCRs in experimentally-verified NS1/NS3-reactive memory cells isolated on day 120.

ELISPOT analysis of the day 120 post vaccination sample demonstrated that NS1 and NS3-reactive cells were the most abundant TAK-003 elicited memory cells present at this time point, but we are not attempting the claim that all NS1 and NS3 reactive T cells present at day 14 post vaccination are likely to become long-lived memory cells, nor that NS1- and NS3-reactive CD8⁺ T cells are dominant at this time point. The relative abundance of NS1- and or NS3-reactive CD8⁺ T cells at day 14 post TAK-003 vaccination does not factor into our analysis, nor does the amount of vaccine-derived antigen, only the relative abundance of NS1- and/or NS3-reactive CD8⁺ T cells in the stable memory pool 120 days post vaccination.

We are not attempting to make any claim about the relative abundance and specificity of all DENV-reactive CD8⁺ T cells present on day 14 post vaccination, rather that the CD8⁺ T cells which express the same TCR as memory CD8⁺ T cells can be identified and form a unique population. As we are highlighting only activated cells at day 14 that express the same TCR as the most abundant memory cells found 120 days post vaccination, we feel very confident that we are accurate in our assertion that we are capturing and identifying memory precursor CD8⁺ T cells in our analysis.

Reviewers' Comments:

Reviewer #1:

Remarks to the Author:

The authors have addressed all my comments, and left few minor items

May I suggest to adapt the following

1) If possible I would suggest to move in the main Suppl. Fig 4 as it clearly shows the difference with the CD38- population.

2) Now the paper has a clear message regarding the transcriptomic profile of HLADR+ CD38+ cells and the heterogeneity therein.

I was indeed reading the new paragraph and a question came to my mind

Recently a scRNA-seq analysis of CD38+ HLA DR+ CD8 T cells was published in Nature Comm on H7N9 Influenza and the gene signature (Wang et al Nat Comm 2018, Clonally diverse CD38+HLA-DR+CD8+ T cells persist during fatal H7N9 disease.). Can the authors comment on the similarity between their vaccination study and a real viral infection, such as flu?

3) There was a typo in L134 "Tome point" should be time point

4) The term "Matrixed elispot" is a bit unclear, perhaps matrix-based ELISPOT may be more accurate

5) The authors have not clearly explained the differential expression analysis performed in cell ranger. What parameters have been used for thresholds? 2-fold? what was used as cut off for the gene expression shown in the tables?

Reviewer #2:

Remarks to the Author:

The authors have done some of the modifications I requested, but they still have one that overstates some aspects of the metabolic data. I will only comment about this.

1) The authors state that they see changes in the amino acid transporter LAT1 by flow cytometry (fig 4). They do not measure LAT1 they measure its chaperone CD98. The text should be modified to say that they measure CD98. This does not necessarily mean LAT1 is expressed as CD98 can partner with other system L transporters and with integrins. In this context the authors in their rebuttal that the metabolites transported by CD98 fuel oxidative phosphorylation. CD98 is not a transporter but it can partner system L amino acid transporters that transport large neutral amino acids. However these do not fuel oxidative phosphorylation.

2) To validate that they were measuring glucose transporters they were asked to do a cold competition assay in their 2NPDG binding experiments. This did not work well and rather than accept that this means that their assay is not measuring a glucose transporter they claim this is an affinity problem. This is tiresome and not good science. If one cannot do the experiment properly then one does not assume that the data are valid ones. Should rather be more questioning. However this could easily be overcome if the authors are open and honest about their data. Eg they should discuss the issues with cold competition in the paper. This would be simple to do. Readers can then make their own mind up

Reviewer #3:

Remarks to the Author:

I read the revision with great interest.

In contrast to the interpretations presented here, what I can conclude from the data presented in the revised manuscript is: after vaccination the authors are identifying a set of highly activated effector cells. These cells, although highly activated based on activated profiles as well as metabolic indicators, completely fail to produce IFN-g when stimulated with peptides.

This phenotype is similar to what has been reported in dengue natural infection in humans, including robust up-regulation of CD71 , massive proliferation and inability to produce IFN-g when stimulated with dengue peptides (see: handele A, Sewatanon J, Gunisetty S, Singla M, Onlamoon N, Akondy RS, Kissick HT, Nayak K, Reddy ES, Kalam H, Kumar D, Verma A, Panda H, Wang S, Angkasekwinai N, Pattanapanyasat K, Chokephaibulkit K, Medigeshi GR, Lodha R, Kabra S, Ahmed R, Murali-Krishna K. Characterization of Human CD8 T Cell Responses in Dengue Virus-Infected Patients from India. J Virol. 2016 Nov 28;90(24):11259-11278).

It is anticipated that some, but not all, of these cells are likely to survive to differentiate into memory cells that would regain ability to produce cytokines. These memory cells will obviously share the TCRs of the initial effector cells.

So, I do not think the authors are identifying "memory precursors".

The appropriate interpretation is that: Dengue vaccination leads to a robust expansion of effector cell populations that fail to produce IFN-g, similar to what has been seen in dengue natural infection, but the memory cells generated from these clonotypes later produce IFN-g.

Reviewer #1

If possible I would suggest to move in the main Suppl. Fig 4 as it clearly shows the difference with the CD38-population.

While we agree with the reviewer that the data shown in Supplemental Figure 4 adds significant value to the manuscript, moving this figure to the main body of the manuscript would exceed the maximum number of figures/tables allowed for our submission

Recently a scRNA-seq analysis of CD38+ HLA DR+ CD8 T cells was published in Nature Comm on H7N9 Influenza and the gene signature (Wang et al Nat Comm 2018, Clonally diverse CD38+HLA-DR+CD8+ T cells persist during fatal H7N9 disease.).Can the authors comment on the similarity between their vaccination study and a real viral infection, such as flu?

We are very familiar with the manuscript mentioned by the reviewer, as it has provided inspiration for some of our own work presented in this manuscript. However, we observe minimal overlap between the differentially-expressed gene products highlighted in the Wang et al study and those emphasized in our work. This is attributable to the fact that Wang et al were assessing differential gene expression between putatively pathogenic and non-pathogenic populations of influenza-reactive CD38⁺ HLA-DR⁺ CD8⁺ T cells, while we are assessing differential gene expression within a population of CD38⁺ HLA-DR⁺ CD8⁺ T cells in individual subjects. The gene signatures presented in the Wang et al manuscript includes many differentially-expressed interferon-associated factors, which we do not observe in our dataset.

However, many of the gene products and surface markers that we have highlighted in our study associated with T cell proliferation and effector function have been observed to be expressed in CD8⁺ T cells responding to natural viral infections. We have updated our manuscript to highlight these conserved transcriptional signatures.

There was a typo in L134 "Tome point" should be time point

We have corrected this typo in the revised manuscript

The term "Matrixed elispot" is a bit unclear, perhaps matrix-based ELISPOT may be more accurate

We agree with the reviewer that our description of this technique was confusing. We have clarified the description of the matrix-based ELISPOT in our revised manuscript.

**The authors has not clearly explained the differential expression analysis performed in cell ranger
What parameter have been used for thresholds? 2-fold? what was used as cut off for the gene expression showed in the tables?**

Differential gene expression in CellRanger is calculated using a sSeq negative binomial exact test (Yu, Hubber, & Vitek, 2013), paired with the edgeR fast asymptotic beta test (Robinson and Smyth, 2007) for samples with a large number of counts. For each unique cluster, the algorithm is run on that cluster relative to all other clusters, generating a list of genes that are differentially expressed in the cluster of interest relative to all other cells in the sample.

The genes presented in Table 3, Supplemental Table 2, and Supplemental Table 4 are the top 20 differentially expressed gene in each cluster, first ranked by p-Value (using the Benhamini-Hochberg procedure to control for the False Discovery Rate), then ranked by fold-change. We did not apply any threshold for significance in displaying the data, opting to provide all gene information and statistical values.

However, we agree with the reviewer that this information should have been made clearer, and have modified the pertinent tables by **1)** adding a fold-change column in addition to the p-value column to clarify the ranking and selection criteria used to generate the table, and by **2)** placing an asterisk next to genes which do not reach a Benhamini-

Hochberg adjusted p-Value of 0.05. We have updated the methods section to explain in more detail the selection methodology used for generating the differential gene expression data and the aforementioned tables.

Reviewer #2

The authors state that they see changes in the amino acid transporter LAT1 by flow cytometry (fig 4). They do not measure LAT1 they measure its chaperone CD98. The text should be modified to say that they measure CD98. This does not necessarily mean LAT1 is expressed as CD98 can partner with other system L transporters and with integrins. In this context the authors in their rebuttal that the metabolites transported by CD98 fuel oxidative phosphorylation. CD98 is not a transporter but it can partner system L amino acid transporters that transport large neutral amino acids. However these do not fuel oxidative phosphorylation.

We wish to thank the reviewer for pointing out our error in terminology. Our goal was to emphasize and clarify the functionality of the surface markers we were assessing in our study, but in doing so we obfuscated the true identity of the marker being quantified. We have changed all references to LAT1 expression back to CD98 in our manuscript, as was presented in the initial submission.

In addition, in our previous response to the Reviewer we were not attempting to claim that the large neutral amino acids transported by CD98/LAT-1 (valine, leucine, isoleucine, tryptophan and tyrosine) directly fuel mitochondrial oxidative phosphorylation. Rather, after enzymatic processing these substrates can be fed into the TCA cycle as acetyl CoA, Succinyl CoA, or Fumarate, which can subsequently provide reducing potential to fuel mitochondrial oxidative phosphorylation.

To validate that they were measuring glucose transporters they were asked to do a cold competition assay in their 2NPDG binding experiments. This did not work well and rather than accept that this means that their assay is not measuring a glucose transporter they claim this is an affinity problem. This is tiresome and not good science. If one cannot do the experiment properly then one does not assume that the data are valid one. Should rather be more questioning. However this could easily be overcome if the authors are open and honest about their data. Eg they should discuss the issues with cold competition in the paper. This would be simple to do. Readers can then make their own mind up

We appreciate the point being made by the reviewer, and have added language to the revised manuscript describing the results of the competition assay without attempting to provide un-supported context.

Reviewer #3

I read the revision with great interest.

In contrast to the interpretations presented here, what I can conclude from the data presented in the revised manuscript is: after vaccination the authors are identifying a set of highly activated effector cells. These cells, although highly activated based on activated profiles as well as metabolic indicators, completely fail to produce IFN-g when stimulated with peptides.

This phenotype is similar to what has been reported in dengue natural infection in humans, including robust up-regulation of CD71 , massive proliferation and inability to produce IFN-g when stimulated with dengue peptides

It is anticipated that some, but not all, of these cells are likely to survive to differentiate into memory cells that would regain ability to produce cytokines. These memory cells will obviously share the TCRs of the initial effector cells.

So, I do not think the authors are identifying "memory precursors".

The appropriate interpretation is that: Dengue vaccination leads to a robust expansion of effector cell populations that fail to produce IFN-g, similar to what has been seen in dengue natural infection, but the memory cells generated from these clonotypes later produce IFN-g.

We wish to thank the reviewer for highlighting an important conclusion that can be drawn from our study that was not sufficiently emphasized in our previous submission.

We emphatically agree with the reviewer that the majority of the NS1/NS3-reactive memory CD8⁺ T cells that we are able to identify 120 days post vaccination are arising from a robustly expanded pool of effector cells that are incapable of producing IFN- γ upon ex-vivo stimulation. We do not believe that highly activated effector cells and memory precursor cells are categorically incompatible descriptions. Rather, as emphasized by the reviewer, our data suggests that a significant fraction of memory CD8⁺ T cells generated in response to this vaccine product do in fact clonally overlap with a population of highly activated effector cells shortly after vaccination. Our use of the term “memory precursor” to define these cells is not an attempt to downplay their effector phenotype, but simply describing that a fraction of these activated cells are the clonally-linked precursors to memory cells present 120 days post vaccination.

This observation does not preclude the possibility that some memory CD8⁺ T cells generated by this vaccine product may be derived from CD8⁺ T cells clones that do not initially exhibit a robust effector profile, but our data do suggest that the majority of the DENV-reactive memory CD8⁺ T cells generated in response to this vaccine product do clonally overlap with the effector pool.

We have added additional language to our discussion emphasizing these conclusions.